# Large-scale conformal synthesis of one-dimensional MAX phases

Yuting Li[1,2], Haoran Kong [1,2], Jin Yan [1,2], Qinhuan Wang [1], Xiang Liu[1,2], Mingxue Xiang [1] & Yu Wang [1] ✉

MAX phases, a unique class of layered ternary compounds, along with their two-dimensional derivatives, MXenes, have drawn considerable attention in many fields. Notably, their one-dimensional (1D) counterpart exhibits more distinct properties and enhanced assemblability for broader applications. We propose a conformal synthetic route for 1D-MAX phases fabrication by integrating additional atoms into nanofibers template within a molten salt environment, enabling in-situ crystalline transformation. Several 1D-MAX phases are successfully synthesized on a large scale. Demonstrating its potential, a copper-based layer-by-layer composites containing 1% by volume of 1D-$Ti_2AlC$ reinforced phase achieves an impressive 98 IACS% conductivity and a friction coefficient of 0.08, while maintaining mechanical properties comparable to other Cu-MAX phase composites, making it suitable for advanced industrial areas. This strategy may promise opportunities for the fabrication of various 1D-MAX phases.

Nano-sized materials, especially one-dimensional (1D) nanofibers or nanotubes, have attracted significant attention with their unique properties and potential as building blocks for multifunctional materials[1,2]. Their size effect provides remarkable changes in the properties such as smaller diffusion distances, electron confinement effects and enhanced surface properties[2–4]. Hence, the investigation of 1D-nanomaterials may bridge between nanoworld and macroworld by assembling into higher dimensional arrays[5]. Transition metal carbides/nitrides are currently undergoing a resurgence in popularity owing to their favorable properties including remarkable mechanical properties, high stability, and versatility[6,7]. Among them, 3D-MAX phases[8–10] and derivative 2D-MXenes[11,12] exhibiting outstanding application potential has unquestionably catalyzed the ongoing surge in synthesis research. $M_{n+1}AX_n$ phases, commonly known as MAX phases, are a unique class of ternary layered compounds that exhibit combination of properties from both metals and ceramics. Here, "M" represents early transition metal, "A" represents A-group element, "X" is carbon and/or nitrogen, and "n" is typically 1-4[8–10]. Predictably, the counterpart 1D-MAX phases would possess remarkable and distinct properties, and there have been a few efforts for synthesizing such materials in the last decade. Recently, MAX phases with short-range nanofiber morphologies were successfully prepared using carbon nanotubes as carbon source, while their bulk form was clustered powder[13]. Carbon fibers with MAX phases coating were investigated[14–16] while the purity was not ideal. MAX phases hollow rods were also produced using carbon fibers[17], but they were micro-sized and also had low purity. Therefore, the synthesis of 1D-MAX phases with perfectly long-range nanofiber morphology is challenging, their preparation is desirable and significant.

Herein, we propose a conformal strategy for synthesizing 1D-MAX phases with templates assistance in molten salt (MS) environment. The superiority of MS method, with high efficiency and lower consumption[18], has already been demonstrated in the synthesis MAX phases[8,13,19], MXenes[20,21], chalcogenides[22] and so on. More importantly, it realizes conformality of target product effectively given the different solubilities of reactants in molten salts[18]. Although the detailed molecular-level reactions in ionic liquid molten salts remain not fully understood, theoretical calculations suggest that metal atoms dissolve more readily in molten salts than in solid-state reactions[23]. Thus, by introducing additional selected metal elements into 1D-templates in MS environment, which provides a strong polarizing force that destabilizes chemical bonds to facilitate chemical reactions, 1D-MAX phases are nucleated through atomic diffusion reaction in situ[24]. The

[1]State Key Laboratory of Mesoscience and Engineering, Institute of Process Engineering, Chinese Academy of Sciences, Beijing 100190, P. R. China. [2]School of Chemical Engineering, University of Chinese Academy of Sciences, Beijing 100049, P. R. China. ✉e-mail: wyu@ipe.ac.cn

high solubility of the 1D-template in MS, exemplified by TiC nanofibers in this work, ensures the preservation of their nanofibrous morphology. The diffusion of various atoms through the efficient defects channels of TiC nanofibers with lower defect energies, which results in atomic rearrangement and crystalline reconstruction[25]. Ultimately, a series of TiC-based MAX phase nanofibers were prepared in this work.

The conformal synthesized 1D-MAX phases not only preserve the properties of bulk MAX phases, their assembled 2D nanofiber membranes also possess exceptional flexibility and mouldability, which are not revealed by their bulk counterparts[26]. In this work, controllable and large-scale 1D nanofiber membranes are fabricated by taking 1D-Ti2AlC as a representative example. The designed aligned $Ti_2AlC$ nanofibers membranes applied as reinforced phase are combined with copper foils to create with copper foils to prepare a novel layer-by-layer Cu-based composite architecture via the hot-press process. The resulting Cu-based composites exhibit high conductivity and enhanced mechanical properties given the presence of 1D-$Ti_2AlC$, which achieving a very low volume content (1%) and exceptional properties (98 IACS% conductivity and 3-fold increased tensile strength compared with pure Cu). This strategy enables opportunities for synthesizing a wide variety 1D-MAX phases and investigating their properties and potential applications.

## Results

### Conformal synthesis of one-dimensional $Ti_2AlC$

One-dimensional $Ti_2AlC$ was synthesized first, due to its recognition as a commonly used functional ceramic material for its excellent physiochemical properties[27]. The preparation strategy and synthetic procedure were illustrated (Supplementary Fig. 1). Binary transition carbides TiC nanofiber as 1D-template was prepared through designed electrospinning and carbothermal reduction methods (Supplementary Fig. 2, 3), which correspond to the first two steps of the conformal synthetic. Then 1D-TiC was embedded in mixed Ti, Al powder and eutectic NaCl/KCl at 900 °C in vacuum for two hours. During the reaction, more carbon vacancies were presented in the crystalline structure of 1D-TiC with Ti, Al atoms diffusion and elevating temperature, resulting in the formation of $TiC_x$ ($0.5 \leq x \leq 1$). Simultaneously, Ti and Al metal melted and infiltrated into 1D-$TiC_x$ under the strong polarizing force provided by molten salt ionic liquid[18]. The well-ordered $Ti_2AlC$ crystals were finally formed in situ after crystalline reconstruction from TiC to $(TiAl)C_x$ solid solution (Fig. 1a). The morphology of obtained 1D-$Ti_2AlC$ was consistent with 1D-TiC template, confirming the validity of conformal synthesis. More details of underlying mechanism will be discussed later.

The X-ray diffraction pattern (XRD) and Rietveld refinement results (Fig. 1b) verified the formation of a hexagonal crystal structure (P63/mmc) of $Ti_2AlC$ MAX phase with the lattice parameters $a = 0.3025$ nm and $c = 1.3648$ nm (Supplementary table 2). The purity of the 1D-$Ti_2AlC$ was above 95% according to the refinement results, although a little amount of TiC impurity was found for its role as the template. The Raman spectra also evidenced the purity of $Ti_2AlC$[27] (Supplementary Fig. 4). Noting that the $Ti_2AlC$ nanofiber membranes were scalably produced by conformal synthesis (Fig. 1c). Moreover, the assembled nanofiber membranes exhibited good flexibility and free-standing properties (Supplementary Fig. 5). As Ti and Al had much higher solubilities in the molten salt compared to TiC, the final product retained the long-range ordered nanofiber morphology with relatively uniform diameters of about 150 nm (Supplementary Fig. 6), energy-dispersive spectroscopy (EDS) analysis conducted via scanning electron microscopy (SEM) revealed a homogeneous distribution of Ti and Al elements across the nanofibers (Fig. 1d), which indicated the reaction was template induced synthesis[18]. The elements mapping from SEM-EDS showed approximate ratio of 2: 1: 1 with Ti: Al: C, and a little oxygen impurity, which is a common phenomenon during synthesis and detection (Supplementary Fig. 7-9). As mentioned before, the

morphology of $Ti_2AlC$ nanofiber was determined by 1D-TiC templates, whose structure could be adjusted through a designed electrospinning process (Supplementary Fig. 10 and Methods). The high-resolution transmission electron microscopy (HRTEM) images (Fig. 1f, g) along with their Fast Fourier Transform (FFT) mode (Supplementary Fig. 11) displayed the typical layered crystal structure of $Ti_2AlC$ with lattice fringe spacing of 1.36 nm in the selected area shown in Fig. 1e, and the corresponding selective area diffraction (SAED) patterns of 1D-$Ti_2AlC$ were shown in supplementary Fig. 12. An atom-resolved scanning transmission electron microscopy (STEM) image of $Ti_2AlC$ nanofiber (Fig. 1g) clearly revealed the alternating stacking sequences of two Ti atom layers and one Al atom layer. The center-to-center interlayer distance of $1.365 \pm 0.05$ nm calculated from STEM images (Supplementary Fig. 13) was highly consistent with the refinement results above. Based on the characterization results, we could claim that 1D-$Ti_2AlC$ with long-range fibrous morphology was realized compared with other works (Supplementary table 3). The synthesized $Ti_2AlC$ nanofibers retained the physical and electrical properties of bulk $Ti_2AlC$ (Supplementary Fig. 14), and the entangled nanofibers (Supplementary Fig. 15) ensure its flexibility while absent in their bulk counterpart. The modulus of single $Ti_2AlC$ nanofiber was determined by AFM test, the DMT curve revealed the average elastic modulus of 1D-$Ti_2AlC$ was 175 GPa (Supplementary Fig. 11a, b), which was considerable compared with bulk $Ti_2AlC$. One of a classic application of MAX phases is high temperature protective coatings[19], thus the oxidation behavior of 1D-$Ti_2AlC$ was investigated (Supplementary Fig. 11c). The increased weight of samples from 400 °C to 1000 °C suggested Ti and Al elements were oxidized[19]. Compared to bulk $Ti_2AlC$, 1D-$Ti_2AlC$ own lower oxidation activation energy due to its higher surface-to-volume ratio. The unique atomic structure determines that MAX phases have properties of ceramics and metals, thus they exhibit metallic conductivity. Apparently, 1D-$Ti_2AlC$ had higher electric conductivity than 1D-TiC, while lower than bulk $Ti_2AlC$ due to its defective morphologies (Supplementary Fig. 11d).

### Formation mechanisms of 1D-$Ti_2AlC$

In this part, the formation mechanism of 1D-$Ti_2AlC$ and the crystalline reconstruction in conformal synthesis were emphasized, which corresponds to the third step of conformal synthesis. The in-situ crystalline evolution from TiC to $Ti_2AlC$ in nanofibers was analyzed and summarized (Fig. 2a). It is well known that TiC has the highest melting point of 3140 °C and lowest solubility in molten salt compared to Ti and Al metal[13,28], thus serving as the structural template for 1D-$Ti_2AlC$ synthesis. Noting that the carefully controlled ratio of TiP (Ti source) and PVP (C source) in the spinning solution, along with the high reduction, temperature led to the formation of 1D-$TiC_x$[29]. The resulting high density of defects from dislocations, stacking faults and carbon vacancies in $TiC_x$ crystals (Supplementary Fig. 16 and Supplementary Notes) allowed the additional Ti and Al intermetallic compounds to readily diffuse in with lower defects energy[25] to form $(TiAl)C_x$ solid solution, which was reconstructed to $Ti_2AlC$ activated by high temperature. The HRTEM image displayed multiple grains of the selected area in incompletely reacted 1D-$Ti_2AlC$ (synthesis temperature 900 °C and dwell time 0.5 hours), their structures were further confirmed by FFT images (Fig. 2b). The defective $TiC_x$ grains in the right area consisted of randomly distributed nanotwin domains parallel to $(111)_{TiC}$, creating more carbon vacancies (Supplementary Fig. 17) by the segregation of Al, Ti atoms to twin boundaries[30]. The generated defects and carbon vacancies could act as diffusion channel for Al atoms and lower its immigration energy[25,31]. Meanwhile, the gamma phase TiAl intermetallic in left also provided a fast diffusion channel for carbon atoms through twin boundary formed by dislocations and deformation created during synthesis[32]. After the quick nucleation of $(TiAl)C_x$ (Supplementary Fig. 18), the ordered $Ti_2AlC$ crystalline subsequently appeared between TiC and TiAl areas through in-situ evolution

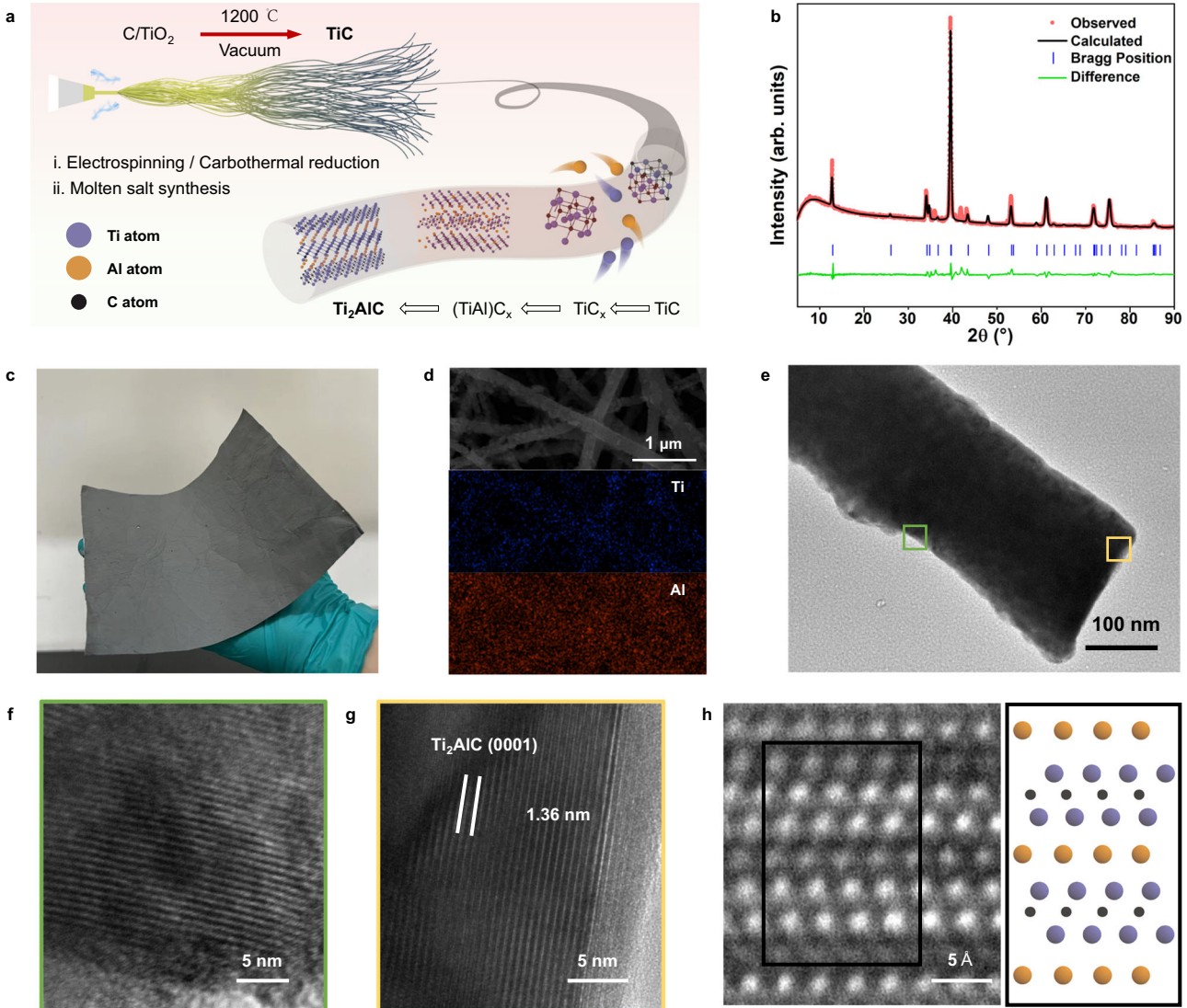

**Fig. 1 | Synthesis and characterization of 1D-Ti$_2$AlC MAX phase. a** Schematic illustration of conformal synthesis and the in-situ crystalline evolution. **b** Rietveld refinement of the XRD pattern of synthesized 1D-Ti$_2$AlC MAX, where the Bragg positions correspond to Ti$_2$AlC as identified by PDF card #29-0095. **c** Photograph of large-scale Ti$_2$AlC nanofiber membranes. **d** SEM image of Ti$_2$AlC nanofibers membranes and the EDS mapping for Ti, and Al elements. **e** TEM image of single Ti$_2$AlC nanofiber shows the long-ranged fibrous morphologies. TEM images of **f**, and **g**, are the enlargements in **e**, along with **h**, HRTEM image of atom positions representing the typical layered structure of Ti$_2$AlC along the [11$\bar{2}$0] axis. Source data are provided as a Source Data file.

(Fig. 2c). The TEM images at different reaction temperatures of incompletely reacted 1D-Ti$_2$AlC demonstrated the similar multiple grain areas (Supplementary Fig. 19), suggested the essence of atomic diffusion for conformal synthesis of 1D-Ti$_2$AlC.

Macroscopically, the formation of 1D-Ti$_2$AlC was accomplished by thermodynamically favored reactions in the molten salt environment. The XRD results at different temperatures and dwell time presented four products involved during synthesis. The intermetallic TiAl$_3$, Ti$_3$Al and TiAl as byproducts were found initially at 800 °C and also observed at higher temperature. They were formed due to excess liquid Al tending to adhere to Ti particles in localized areas with the assistance of salt ions[28,33], according to Eqs. (1), (2) and (3). The main byproduct, TiAl, then reacted with TiC$_x$ to form Ti$_2$AlC through contact and diffusion as discussed previously, according to reaction (4).

$$Ti + 3Al(l) \rightarrow TiAl_3 \tag{1}$$

$$3Ti + Al(l) \rightarrow Ti_3Al \tag{2}$$

$$Ti + Al(l) \rightarrow TiAl \tag{3}$$

$$TiC + TiAl \rightarrow Ti_2AlC \tag{4}$$

When reaction proceeded further, TiC nanofibers were covered entirely in liquid TiAl phase and continued to produce Ti$_2$AlC until there was no residue owing to the decreased diffusion distance and sufficient contact of reactants in molten salt environment. In this work, the product yield of Ti$_2$AlC was quite considerable at 900 °C, which was a significantly lower temperature than those used in conventional synthesis. The calculated negative ΔG values of related reactions suggesting the thermodynamically favorable nature of conformal synthesis, and Ti$_2$AlC was the most stable phase below 1200 °C, which was a major cause for the high purity (Fig. 2d). Differential scanning calorimetry (DSC) and thermogravimetric (TG) performed on reactants in Ar atmosphere to corroborate the formation process of Ti$_2$AlC (Fig. 2e). There were several stages during the reaction based on the DSC curve[33]. The green region represents the continuing formation of NaCl-

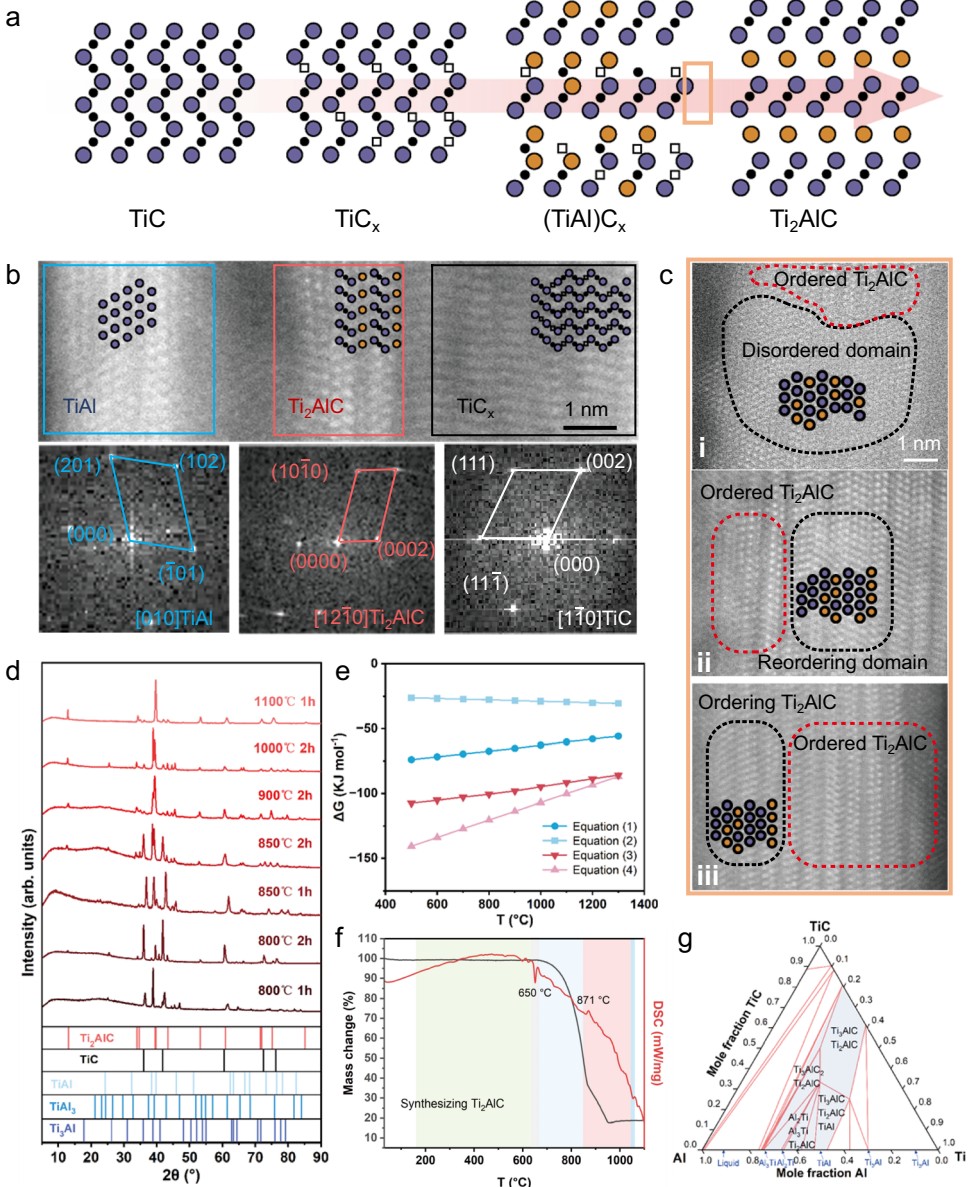

**Fig. 2 | Mechanism of Ti$_2$AlC MAX phase formation. a** Schematics of the in-situ crystalline evolution during the formation of 1D-Ti$_2$AlC. The blue spheres represent Ti atoms, orange spheres represent Al atoms, black spheres represent carbon atoms, and the squares represent carbon vacancies. The symbols of other spheres in this figure are consistent with the description above. **b** TEM image of selected area in the obtained nanofiber from 900 °C with dwell time 0.5 hours presents the relevant products, indicating the Ti$_2$AlC crystals nucleated between TiC$_x$ domain and the diffused TiAl phase with their corresponding FFT images displayed. **c** The HRTEM results of (i) disordered (TiAl)C$_x$, (ii) re-ordering (TiAl)C$_x$, to (iii) disordered Ti$_2$AlC, and ordered Ti$_2$AlC reveals the in-situ reconstruction from (TiAl)C$_x$ to Ti$_2$AlC, which is the detailed process from orange region in **a. d** XRD patterns of the products from 800°C to 1100 °C. **e** Calculated Gibbs free energy of possible reactions from 500 °C to 1300 °C. **f,** DSC and TG curve of a mixture of Ti, Al and TiC nanofibers in the presence of the KCl−NaCl eutectic composition salt under the protection of Ar gas at a rate of 10 °C/min, the colored shades represent the different stages. **g** Calculated phase diagrams of reactants from 900 °C at 0.05 atm pressure, where the blue regions shown the products containing Ti$_2$AlC in this phase diagram. Source data are provided as a Source Data file.

KCl solid solution. The sharp endothermic peak emerging around 653 °C indicated the melt of the salt mixture and Al atoms. The third stage in the blue region represented the formation of Ti-Al inter-metallic compounds. The exothermic peak observed at 871 °C in the next red stage accorded well with the continuous formation of Ti$_2$AlC along with the XRD results above. The sharp endothermic peak at 1018 °C represented the vaporization of molten salts, and the resulting mass loss led to the absence of NaCl and KCl peaks in the XRD results. Additionally, the calculated phase diagram of the Ti−Al−TiC system at 900 °C (Fig. 2g) showed the stable presence of the Ti$_2$AlC phase in a relatively wide range with excessive reactants, therefore, a strict stoichiometric ratio of reactants is not necessary unlike other works[13,27,34].

Hence, the additional metallic elements incorporated were in excess compared to the 1D-TiC templates, thereby enhancing the conversion rate of TiC and promote the yield of 1D-MAX phases. Furthermore, the excess metallic impurities can be readily removed post-reaction via acidification. The surface atomic bonding states of the sample synthesized at 900 °C were characterized and analyzed using XPS. (Supplementary Fig. 20 and Supplementary Table 4). The discussion above verified the feasibility of conformal synthesizing 1D-Ti$_2$AlC at low temperatures in molten salts environment. The synthetic variables were investigated through XRD results and SEM images, including temperatures, molar ratio of reactants, dwell time and HCl treatments (Supplementary Fig. 21, 22 and Supplementary notes).

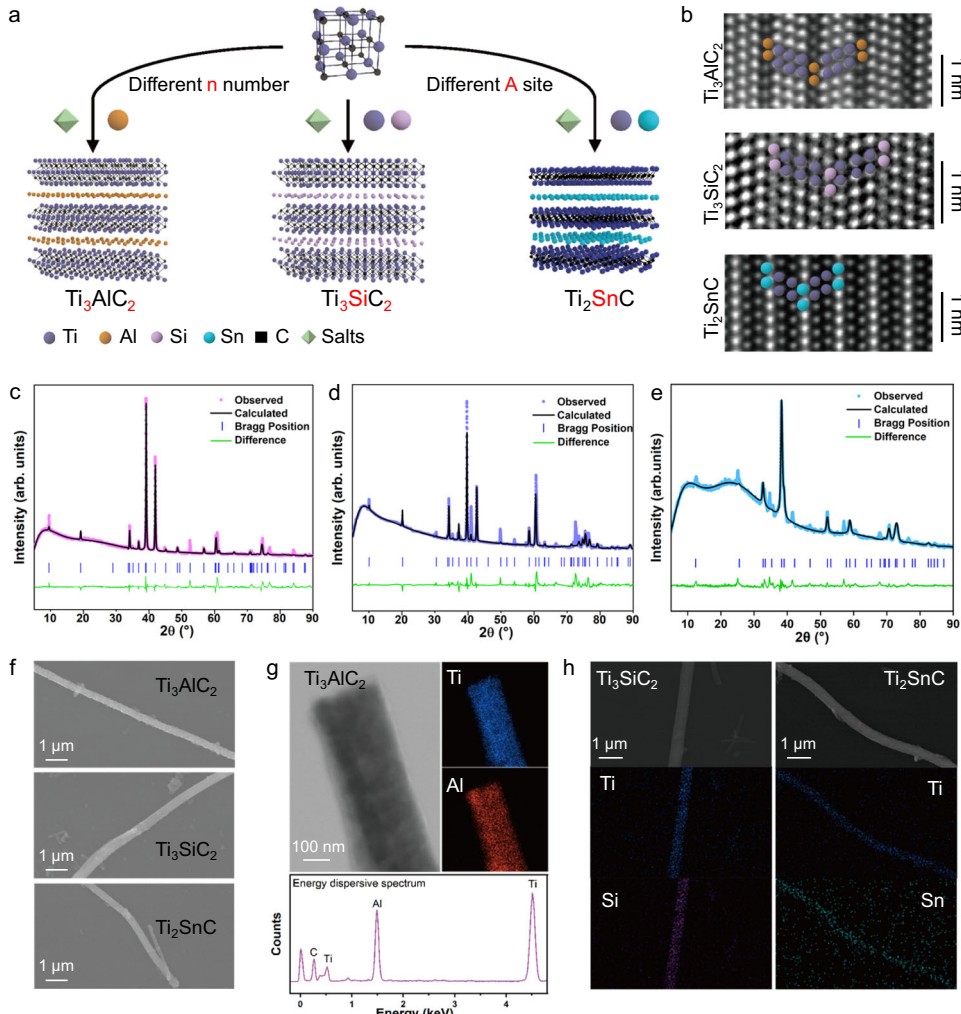

**Fig. 3 | Synthesis of other TiC-based 1D-MAX phases. a** Schematic illustration of conformal synthesis for other 1D-MAX phases with tunable 'n' number and 'A' site atoms in '$M_{n+1}AX_n$'. **b** HRTEM images of $Ti_3AlC_2$, $Ti_3SiC_2$, and $Ti_2SnC$ MAX phases nanofibers. Rietveld refinements of the XRD patterns of **c**, $Ti_3AlC_2$, **d**, $Ti_3SiC_2$, and **e**, $Ti_2SnC$ MAX phase nanofibers, where the corresponding Bragg positions were identified by PDF card #52-0875, 40-1132, and 29-1353. **f**, SEM images of $Ti_3AlC_2$, $Ti_3SiC_2$, and $Ti_2SnC$ MAX phase single nanofibers showing long-ranged morphologies. **g** TEM images of $Ti_3AlC_2$ and corresponding EDS mapping of Ti, Al elements. **h**, TEM images of $Ti_3SiC_2$ and $Ti_2SnC$ MAX phase nanofibers and their corresponding EDS mapping of Ti, Si, and Sn elements. Source data are provided as a Source Data file.

## Conformal synthesis of other TiC-based 1D-MAX phases

To prove the universality of conformal synthesis, other three TiC-based 1D-MAX phases ($Ti_3AlC_2$, $Ti_3SiC_2$, and $Ti_2SnC$) were prepared. It was noteworthy that the stoichiometric number 'n' and 'A' site atoms of 1D-$M_{n+1}AX_n$ phases were tunable by adjusting the species and molar ratio of added elements (Fig. 3a). The corresponding atomic arrangements of three 1D-MAX phases were displayed by STEM images (Fig. 3b), which showed well-ordered stacking sequences of 'M' site atoms and 'A' site atoms. The XRD results, along with Rietveld refinements, confirmed the successful synthesis of highly pure phases (Fig. 3c-e). The detailed synthesis protocols for the various 1D-MAX phases were presented in supplementary materials (Supplementary notes and Supplementary table 1). Similarly, the long-ranged nanofibrous morphologies were retained after conformal synthesis, and the homogeneous distribution of Ti, C, and 'A' sites elements along the nanofibers was observed (Fig. 3f-h). The HRTEM images further demonstrated distinct layered structures with different c parameters along [11$\bar{2}$0] axis (Supplementary Fig. 23-25, 29, 32). Their nanofibrous morphologies and layered structures were shown in supplementary Fig. 35.

The in-situ crystalline transformation in these 1D-MAX phases was also achieved through atomic diffusion and solid solution reconstruction.

For $Ti_3AlC_2$, excess Al atoms diffused in ordered carbon vacancies in $TiC_x$ to form $(TiAl)C_{0.67}$ (Supplementary Fig. 28), while others were mainly formed by interdiffusion between C atoms in $TiC_x$ and 'A' atoms in intermetallic compounds (Supplementary Fig. 31, 34), which were similar to the formation of $Ti_2AlC$. The XRD patterns, combined with DSC results, illustrated the thermodynamically favorable nature of these reactions[35–37] (Supplementary Fig. 26, 27, 30, 31, 33, 34). Owing to the nanoscale nature of templates, the reduced diffusion distance of atoms contributed to the lower reaction temperature than traditional methods[18].

## Design and construction of Cu-$Ti_2AlC$ layer-by-layer composite thin film

Copper is a highly conductive metal; however, it has poor mechanical properties. Thus, alloying it with ceramics reinforcement phases is an efficient method to improve its mechanical properties. MAX phases are suitable reinforcements due to their excellent wettability, machinability and electroconductivity[38]. It was researched that a strong interfacial bond could be formed due to the reaction of Cu and Al when Cu is exposed to $Ti_2AlC$ at a high temperature, where Al from the MAX phase diffused into copper, resulting in the formation of a Cu (Al) solid solution layer and a TiC layer[39]. Therefore, our previous work fabricated a layer-

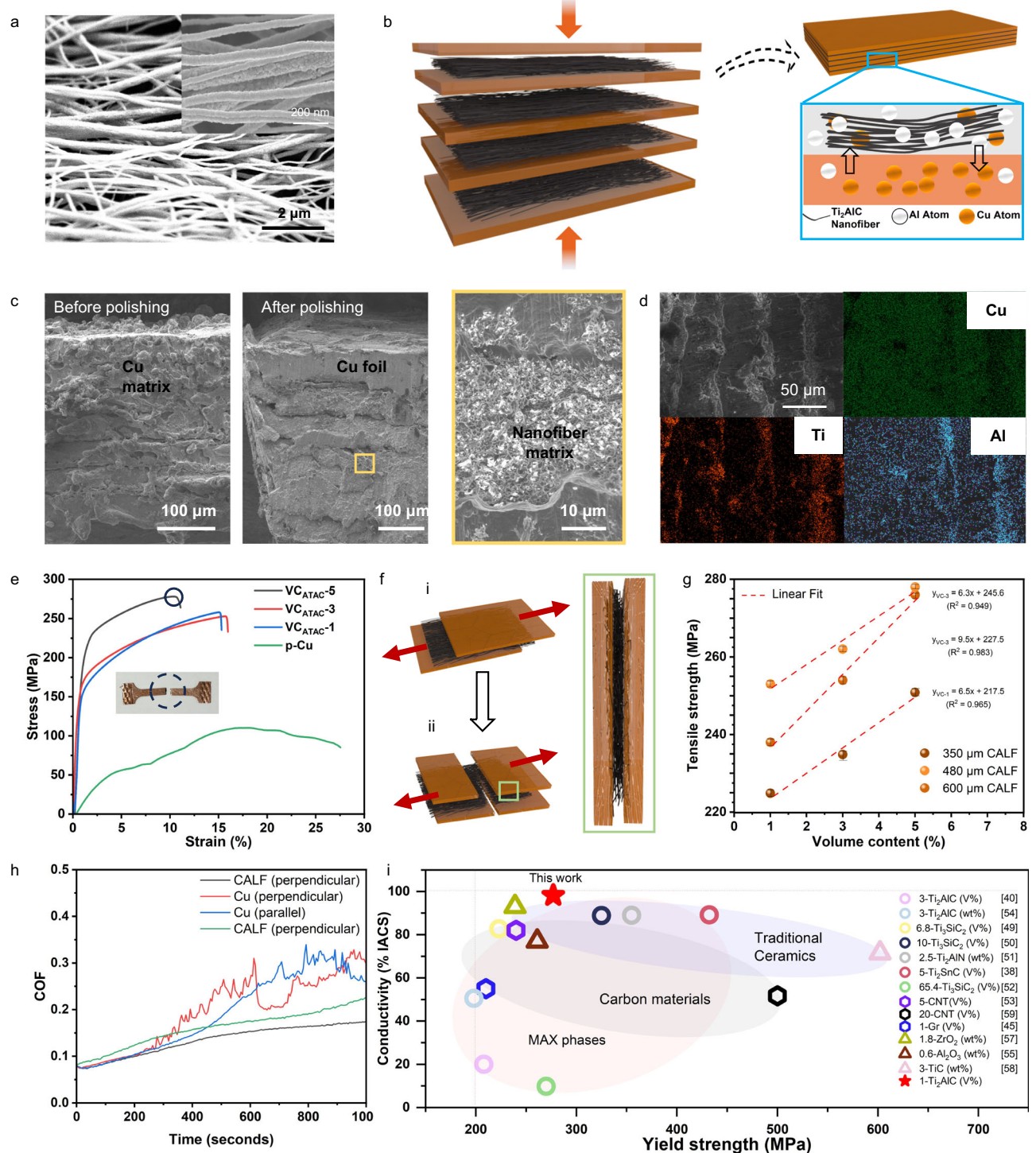

**Fig. 4 | Construction of CALF and investigations of its properties. a** The enlargement of aligned ATAC nanofibers. **b,** Schematic illustration of synthesis mechanism for CALF. **c** SEM images of cross-section of CALF before and after polishing, showed continuous distribution of Cu along the stacking direction, while ATAC nanofiber matrix distributed between Cu layers. **d** Elemental mapping of cross-section of CALF shows the diffusion of Al atom and Cu atom. **e** Tensile stress −strain curves of 600 μm CALFs with different volume content of ATAC nanofibers constructed by stacking different thickness of Cu foils (20 μm, 35 μm, 90 μm and pure Cu foils). **f** Schematic illustration of crack mechanism for CALF. ATAC nanofibers were applied as fiber strengthen phase. **g** The linear increasement of tensile strength of CALF with increased volume content of ATAC. **h,** Coefficient of friction curves conducted from the cross-section of samples along different directions. **i** Comparisons of yield strength and electrical conductivity with other works[38,40,45,51–61]. Source data are provided as a Source Data file.

by-layer Cu-Ti₂AlC laminated bulk material with enhanced properties[40]. In this study, aligned 1D-Ti₂AlC (ATAC) nanofiber membranes were used as the reinforcing phase for its nanofiber film strengthening and dislocation strengthening effects[41,42], to construct a layer-by-layer Cu-based composites for further improvement of properties.

The ATAC nanofiber membranes were first prepared by electrospinning using high-speed rotating collector, followed by the same post-processing as general 1D-MAX phases (Fig. 4a). The nanofibers constituted an orderly arrangement by strong traction force through rotating collector[43](Supplementary Fig. 36). Statistical analysis also

confirmed the high alignment of ATAC nanofibers (Supplementary Fig. 37). Then Cu foils and ATAC nanofiber membranes were stacked layer by layer and hot-pressed at 1000 °C, during which diffusion between Cu and Al atoms occurred at the interfaces (Fig. 4b and Supplementary Fig. 38). The SEM image of the cross=section of obtained Cu-ATAC layered composites films (CALF) showed a significant phenomenon where Cu permeated along the stacking direction, which was attributed to the intrinsic network morphologies of the nanofiber membranes, and the large ratio of thicknesses of Cu foil (90 µm) and nanofiber membranes (10 µm). The continuous distribution of Cu along and perpendicular to the stacking direction ensured the high conductivity with increasing mechanical properties. The layer-by-layer nature of CALF then revealed after polishing, and the 1D-ATAC nanofibrous matrix, which transformed into TiC after the reaction, exhibited a net frame morphology due to interface reaction. This morphology provided more locking points between nanofibers to enhance the binding force (Fig. 4c). The EDS results and the XRD results (Supplementary Fig. 38) of the cross-section indicated that Cu and Al elements are diffused with each other throughout the entire materials (Fig. 4d).

To investigate the mechanical properties of CALF, Cu foils of various sizes were used to prepare films with different thicknesses (0.35, 0.46, and 0.6 mm) and different volume content (1%, 3%, and 5%) of reinforced phase (Supplementary table 5). The tensile-stress results suggested the maximum tensile strength (277 MPa) was three times higher than pure copper (Fig. 4e). The remarkable increase in the mechanical properties were realized by the strong bonding interface between the hard phase layer ATAC and soft phase layer Cu[40]. The fracture surface analysis indicated that CALF composites exhibited both brittle and ductile fracture behaviors. When external force was applied, microcracks were produced in the aligned fibrous matrix first (Supplementary Fig. 39), then the energy stored in crack was released through the plastic deformation of soft layers[40,41]. Due to the layer-by-layer property of CALF, the extension of the cracks was impeded when they encountered the neighboring hard phase layer, thus leading to a better tensile strength[44] (Fig. 4f). It is evident that tensile strength improved linearly with thicker films due to the increased volume contents of reinforced phase (Fig. 4g). Thinner copper foils could incorporate a greater volume fraction of ATAC NFMs, consequently exhibiting superior mechanical properties (Fig. 4e and Supplementary Fig. 40). Also, the nanosized ATAC provided a fine-grained strengthening mechanism[45], where traditional MAX phases had larger grain size. The Vickers hardness and Young's modulus of CALF were 2 times and 6 times higher than those of the pure copper (Supplementary Fig. 41, 42). The tribological tests of CALF illustrated the enhanced antifriction properties (coefficient of friction 0.08 the lowest) compared with pure copper (COF of 0.6) and other Cu-MAX composites[38], and the layer-by-layer structure resulted in the anisotropic properties along and perpendicular to the stacking directions (Fig. 4h and Supplementary Fig. 43, 44). It worth noting the COF of large area of the Cu (Al) surface also possessed superior anti-friction characteristics, suggesting its potential for practical applications. Despite the excellent tribological properties of CALF, its resistivity was also as same order of magnitudes as pure copper[46,47] (Supplementary Fig. 45). More importantly, the amounts of reinforced phase (1 vol%) in this work were much less than those in the previous research, while the mechanical properties were in the same level with other Cu-based composites (Fig. 4i). Meanwhile, the low content of ceramics also preserved the superior conductivity of Cu, hence the resistivity of CALF was much lower than other Cu-based materials. Predictably, the constructed CALF material could be an ideal candidate in thermal engines, spacecrafts, pantographs and other cutting-edge industrial applications due to its outstanding properties[26,48].

The conformal strategy in this work could provide a universal route aimed at producing other 1D-MAX phases, since each step of the conformal strategy is flexible and adaptable. For instance, the template could be other binary transition metal carbides/nitrides composed of 'M', 'A' and 'X' site atoms, or the single-atom template like carbon nanofibers. The complementary process for lacking atoms is also diverse such as physical vapor deposition, electrophoretic deposition, even solid-state reaction. Besides, it was shown that 1D-MAX phases were also capable of being etched into MXenes by HF (Supplementary Fig. 46). The exploration of 1D-MAX phase nanofibers could be easily amenable to further scaling and expansion accordingly. Moreover, the properties and other potential applications of 1D-MAX phases are equally critical for further investigations, as well as the in-depth understanding of dimensional change for this material.

## Methods

### Synthesis of TiC nanofibers templates

Titanium isopropoxide (TiP), poly(vinylpyrrolidone) (PVP, Mw = 1,300,000) were purchased from Aladdin Reagent Co., Ltd., China. Acetic acid (HAc), N, N-dimethylformamide (DMF), chloroform (CF), NaCl (99.99%, ultra-dry), and KCl (≥ 99.95%, ultra-dry) were purchased from Shanghai Maclean Biochemical Technology Co., Ltd. The graphite papers (35 µm, 98%) were purchased from JingLong Special Carbon Co., Ltd.

TiC nanofibers were prepared using sol-gel method combined electrospinning and subsequent carbothermal reduction processes. The spinning sols were prepared by mixing titanium isopropoxide (TiP), N, N-dimethylformamide (DMF), chloroform (CF), polyvinyl pyrrolidone (PVP, MW = 1.3 m), and acetic acid (HAc) with a weight ratio of 3: 1: 9: 1: 2 with strong stirring for 6 hours. The obtained gel was used as the spinning solution. Electrospinning was carried out by applying a fixed 18 kV voltage between an 18 G stainless steel needle connected to a 10 mL syringe filled with spinning solution and a rotating drum covered by graphite paper as a collector for 2 h with a gel fed ratio of 2 mL h$^{-1}$. The as-prepared nanofibers were firstly preheated for 3 h at 220 °C in air with ramping rate of 1 °C min$^{-1}$, then collected after carbothermal reduction process at various temperatures (1100, 1200, and 1300 °C) for 1 h with heating rate of 10 °C min$^{-1}$ in vacuum in a tube furnace. Although higher temperature could increase the crystallinity of TiC nanofibers, the flexibility of nanofibers was decreased. Therefore, TiC nanofibers at 1200 °C considered as the ideal templates in this study.

The morphologies of nanofibers and its assembled nanofiber membranes could be customized by changing electrospinning parameters. Nanofibers with larger diameters were obtained by decreasing voltage to 15 kV, since the charge density of spinning solution decreased lead to the increased jet radius. Porous nanofibers were produced by blending above spinning solution and polytetrafluoroethylene (PTFE) suspension with a weight ratio of 1: 1. PTFE nanoparticles were decomposed to porous structure during heating process, thus the porous nanofibers would be obtained. By increasing the spinning time to 4 h, the nanofiber membranes possessed a larger thickness. The following post process was the same[49,50].

### Synthesis of 1D-Ti$_2$AlC MAX phase

Ti powder (99%, particle size <5 µm), Al powder (99.5%, particle size <5 µm), Si powder (99.5%, particle size <5 µm), Sn powder (99.5%, particle size <5 µm) were purchased from General Research Institute for Nonferrous Metals, China. HCl solution (37 wt %) was purchased from China Shanghai Sinopharm Chemical Reagent Co., Ltd.

Ti$_2$AlC MAX phase nanofibers were obtained by template-assisted conformal synthesis in molten salts environment using TiC nanofibers precursors. Firstly, the eutectic molten salts were prepared by mixing NaCl and KCl with a molar ratio of 1: 1 and ground for 5 min. Then Ti and Al powder were added in the salts with a molar ratio of 1: 1.2: 4: 4 (Ti: Al: NaCl: KCl). The obtained powder mixture was ground further for half an hour and dried at 70 °C in a vacuum for 12 hours. Afterwards, the mixed powders were evenly covered onto both the upper

and lower sides of the TiC nanofibers in a proportion of 125 mg cm$^{-2}$ in a crucible with a tightly-sealed lid, then heated in a tube furnace at 900 °C with a dwell time of 2 h in a vacuum. Noting that the tube furnace was cleaned before heating with pure Ar gas for three times to remove air in case of oxidation of reactants. The heating rate was 10 °C min$^{-1}$. After naturally cooling down to the room temperature, the obtained nanofibers were immersed into 6 M HCl for 6 hours and transferred into deionized water for another 6 hours to remove any possible residues and the excessive reactants. Finally, the 1D-Ti$_2$AlC were retrieved after drying at 70 °C for 20 mins in an oven.

### Synthesis of 1D-Ti$_3$AlC$_2$ MAX phase

Ti$_3$AlC$_2$ MAX phase nanofibers were prepared with a similar process. Specifically, Al metal powder were added into NaCl and KCl eutectic salts with a ratio of 1: 4: 4 and ground for half an hour. The powder mixture was dried at 70 °C in a vacuum for 12 hours. Then the mixed powder was covered onto both sides of the TiC nanofibers with 60 mg cm$^{-2}$. The reaction temperature was 1000 °C and dwell time was one hour. The heating rate was 10 °C min$^{-1}$. The Ti$_3$AlC$_2$ nanofibers were collected under the same post-processing procedure.

### Synthesis of 1D-Ti$_3$SiC$_2$ MAX phase

Ti$_3$SiC$_2$ MAX phase nanofibers were obtained by embedding TiC nanofibers in mixture powder and heating at 1100 °C for 1 h in vaccum with the heating rate of 10 °C min$^{-1}$. The mixture powder contained Si metal powder, Ti metal powder and NaCl + KCl eutectic salts with a molar ratio of 1.2: 1: 4: 4, plus 0.2 molar ratio of extra Al added as catalyst, and ground for half hour. The powder mixture covered with 100 mg cm$^{-2}$. The post-processing procedure was same.

### Synthesis of 1D-Ti$_2$SnC MAX phase

Ti$_2$SnC MAX phase nanofibers were obtained by replacing the Al into Sn for the powder mixture. The molar ratio of Ti powder, Sn powder, and NaCl/KCl eutectic salts is 1.2: 1: 4: 4. With the similar procedure, the synthesis temperature was 850 °C and the corresponding dwell time was 2 hours. The post-processing procedure was same.

### Synthesis of aligned 1D-Ti$_2$AlC nanofibers

Aligned Ti$_2$AlC nanofibers were prepared using similar procedure while the rotating speed of drum collector was 1500 rpm and spinning time was 30 min for electrospinning process. The post processing was same as general nanofibers, including carbothermal reduction process to obtain TiC nanofibers, and subsequent template-assisted conformal synthesis for aligned 1D-Ti$_2$AlC. The obtained nanofiber membranes were also post-processed.

### Construction of Cu-aligned Ti$_2$AlC nanofibers layered composites films

The copper foil with thicknesses of 20 μm, 35 μm and 90 μm (TU2, 98%) were purchased from MingYu metal materials Co., Ltd. Copper films with different sizes (20 μm, 35 μm, and 90 μm) were cut into 3.5 cm × 2.5 cm pieces, and cleaned by diluted HCl and deionized water three times. The aligned Ti$_2$AlC nanofiber membranes (ATAC NFMs) were cut into the same size. Then Cu foils and ATAC NFMs were periodically alternating stacked with certain layers (Supplementary Table 5), and transferred into a hot-press furnace. The final products were obtained after hot-pressed sintering at 1000 °C for 1 h with a uniaxial pressure of 25 MPa. The obtained Cu-ATAC layered composites films (CALFs) with different thicknesses were determined by the assembled layers and the sizes of Cu foils. In this work, CALFs with thicknesses of 0.35 mm, 0.48 mm, and 0.6 mm were used and characterized with certain volume contents of the reinforcement phase of ATAC (1%, 3%, and 5%).

### X-ray diffraction (XRD)

The diffraction patterns in Bragg-Brentano geometry were obtained using a Rigaku SmartLab 9 kW diffractometer using Cu Kα radiation (l = 0.15406 nm) operating at 40 kV and 15 mA at a scan rate of 2 ° min$^{-1}$. XRD full pattern fitting (Rietveld refinement) was performed using GSAS-II software. The 1D-MAX phase samples were assumed to contain two phases: a major MAX phase (space group *P*-63/*mmc*) and a trace amount of TiC (space group *Fm*-3*m*).

### Scanning electron microscopy-energy dispersive X-ray spectroscopy (SEM-EDS)

The surface morphologies were characterized using a field-emission scanning electron microscope (Hitachi SU8020, Japan) with an equipped attached dispersion micro-analysis of energy detectors. The accelerating voltage was set to 15 kV.

### Transmission electron microscopy (TEM) and Scanning transmission electron microscopy (STEM)

Microstructural analyses of samples were carried out using a JEM-F200 (cyro) transmission electron microscope. High-angle annular dark field (HAADF) scanning transmission electron microscope (STEM) for atomic-resolution characterization was conducted using JEOL JEM-ARM200F microscope incorporated with a spherical aberration correction system for STEM. STEM Energy-dispersive X-ray spectroscopy (EDS) mapping was performed using a 100 mm$^2$ JEOL Centurio SDD EDS detector. The samples were prepared by drop-casting nanofibers suspended in ethanol onto a 3 nm holey-carbon covered TEM grid and dried over 2 h. The TEM grids were then loaded onto a plasma-cleaned Fischione vacuum transfer holder and inserted into the microscope column without atmospheric exposure.

### Raman spectroscopy

Raman spectra were obtained with a Renishaw inVia Raman microscope. The samples were excited using a 532 nm laser with a power of 1 mW.

### Atomic force microscopy (AFM)

AFM data was obtained with a Bruker Dimension Icon Atomic Force Microscopy using its tapping mode. The Derjaguin-Müller-Toporov (DMT) mode was also recorded to investigate the modulus of nanofibers. AFM samples were prepared by drop-casting and drying diluted nanofiber suspension in ethanol on Si wafers.

### X-ray photoelectron spectroscopy (XPS)

XPS analysis of 1D-Ti$_2$AlC was performed on a Thermo Scientific ESCALAB 250Xi spectrometer using a monochromatic Al X-ray source. Ti 2p, Al 2p, C 1 s, and O 1 s high-resolution spectra were collected using an analysis area of 0.3 × 0.7 mm$^2$ and a 20-eV pass energy with a step size of 100 meV. Peak fitting of high-resolution XPS spectra was performed with Avantage software. The binding energy scale for charge correction was assigned by adjusting the C 1 s peak at 284.8 eV.

### Thermogravimetric analyses (TGA) and differential scanning calorimetry (DSC)

To investigate the reaction mechanisms, the TGA and DSC curves of 1D-MAX phase synthesis were performed on a thermogravimetric analyzer instrument (TGA-DSC 3 + , Mettler Toledo, Switzerland) from 25 °C to 1200 °C with a heating rate of 10 °C min$^{-1}$ under flowing air atmosphere. Samples were prepared by mixing 1D-MAX phases and their corresponding powder mixtures with certain molar ratios.

To investigate the oxidation performance of 1D-Ti$_2$AlC, TGA and DSC were carried out in air from 25 °C to 1200 °C with a heating rate of 10 °C min$^{-1}$.

## Calculation of Gibbs free energy and phase diagram

The Gibbs free energy of related reactions during synthesizing 1D-MAX phases were calculated by Factsage software. The values of $\Delta H_r$ (r stands for reactants) and $\Delta S_r$ can be obtained from the database, $\Delta G_r$ is given by:

$$\Delta G_r = \Delta H_r - T\Delta S_r \tag{5}$$

The reaction temperatures were set from 500 °C to 1200 °C, and the molar ratio of reactants Ti, Al, and TiC were set as 1: 1: 1, since the excess reactants will not further to form MAX phases once TiC nanofiber was completely transferred into $Ti_2AlC$.

The phase diagrams were obtained using Factsage software. The thermodynamic parameters were also obtained in the database. The reaction pressure and temperature were set same as real experiments to compute the corresponding phase diagrams.

## Characterizations of CALF

The physical properties of CALF, including density, volume ratio of added ATAC NFs, and electrical properties. The density of samples was measured using Archimedes' method. The weights of samples (M) were measured first, then a cup of water was placed on an electronic balance. Then the sample was suspended in water and the increment on the electronic balance was recorded, which was the mass of the water displaced by the samples (m). The density of sample (D) then could be calculated by the following equation, where the density of water ($\rho$) is $1.0 \, g\,cm^{-3}$.

$$D = \frac{M}{m}\rho = \frac{M}{m}\left(g\,cm^{-3}\right) \tag{6}$$

The results were displayed in Table S5. The volume contents (VC) of the added reinforcing phase 1D-$Ti_2AlC$ were calculated through the following formula,

$$VC = \frac{\frac{m-m_{Cu}}{4.1}}{\frac{m-m_{Cu}}{4.1} + \frac{m_{Cu}}{8.94}} \tag{7}$$

where $m_{Cu}$ was the weight of copper foil used, and 8.94 was the density of copper and 4.1 was the density of $Ti_2AlC$ ($g\,cm^{-3}$). The resistivity was measured via four-probe resistance measurement (RTS-9, Guangzhou Four Probe Technology, China).

## Mechanical properties of CALF

The tensile tests were carried out in a universal testing machine (CMT4105, China) at a strain rate of $1 \times 10^{-4} \, s^{-1}$. The Young's modulus was directly obtained through the tensile-strain curves. The tested samples were cut into dumbbell shapes by wire-electrode-cut methods to satisfy the standards for tensile tests. The hardness was measured by a Vickers hardness tester (THVS-30 Shidai, China) using a static load of 5 kg f and a dwell time of 15 s. Tribological tests were examined on a ball-on-flat tribometer (Bruker UMT3) at the room temperature. In the friction test, the 100Cr6 steel ball with a diameter of 10 mm as the upper counterpart. The surface roughness and hardness of the ball were 0.02 μm and 66 HRC, respectively. The lower specimen was CALF composite its length, width and height were 5, 5 and 3 mm, respectively. For reciprocating sliding, the upper ball slid against the fixed CALF sample with an oscillation frequency of 1 Hz and a stroke length of 1 mm. Each test was performed under a 20 N normal load for 300 s and repeated three times.

## Data availability

Source data are provided with this paper.

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

## Acknowledgements

This work was supported by the National Natural Science Foundation of China (No. 21875256), Beijing Natural Science Foundation (IS23042), the Science and Technology Service Network Initiative of CAS (KFJ-STS-QYZD-2021-14-001), and research fund of State Key Laboratory of Mesoscience and Engineering (MESO-23-A06).

## Author contributions

Yuting Li: Methodology, Data curation, Writing – original draft, Visualization, Investigation. Haoran Kong: Methodology, Visualization. Jin Yan: Resources. Qinhuan Wang: Investigation. Mingxue Xiang: Methodology. Xiang Liu: Investigation. Yu Wang: Conceptualization, Resources, Funding acquisition, Writing – review & editing, Supervision

## Competing interests

The authors declare no competing interests.
