## [Peer Review File · Nature Communications]

Large-scale conformal synthesis of one-dimensional MAX phasesREVIEWER COMMENTS

Reviewer #1 (Remarks to the Author):

The paper by Li et al. is interesting work and should in principle be published. Before doing so however there are a few issues that need to be addressed.

First and foremost the authors need to quantify their XPS spectra. How much of the Ti is bound with C in MAX phase and what fraction is in the oxide state. This is important because in many cases working with MAX phases and MXenes oxidation of the Ti becomes an issue.

In general, I would like to see many more SAD diffraction patterns. For example I would like to see SAD for Fig. S18. The same for Figs. 1f and 1g.

In Fig. 2f the authors claim the large exothermic peak is due to the evaporation of the salt. Normally evaporation is endothermic. What am I missing?

The claim that the composites could be used in many applications in where low friction was important is very debatable. Which surface do the authors propose to use their composites? The large usable area is the top area with is pure Cu, no?

Other problems

The references do not refer to any of the pioneering work by Barsoum on the MAX phases. Also in certain places the authors make claims with no refs. For example, in Fig. S4 and S10e they claim that Raman spectrum agrees with previous work. What previous work? Pls reference. Reference for conductivity of pure Cu?

Reviewer #2 (Remarks to the Author):

Yuting Li et al. provided lateral thinking to fabricate 1D MAX nanofiber, and the research has innovativeness and systematicness. Then, we concerned several questions as follows:

1. Why authors choose to fabricate TiC nanofibers first but not to use commercial carbon nanofibers? Is the reason the existence of amorphous carbon in carbon nanofibers?
2. What are the impurities in MAX nanofibers? If the impurities include ceramic phases like TiC, how did you eliminate them using HCl?
3. What length do the MAX nanofibers extend? And the length-to-diameter of the MAX nanofibers should be given in the text or graph sections.
4. HCl can etch the MAX phase together, how did you avoid this problem?
5. Fig. 4 d doesn't have the scaleplate, thus confusing readers by an unclear scale for the distance of element diffusion. What is important is what reaction happened between Ti₂AlC and Cu. The authors should provide the XRD result of 1-Ti₂AlC/Cu composite. Does anything else like Al₂O₃ exist in the composite because the Al in Ti₂AlC will be easy to combine with O during hot-pressing process.
6. Why did not the Ti₂AlC react entirely to be TiC when composited with Cu at 1000°C?

RESPONSE TO REVIEWERS' COMMENTS

Reply to Reviewer 1

The paper by Li et al. is interesting work and should in principle be published. Before doing so however there are a few issues that need to be addressed.

First and foremost the authors need to quantify their XPS spectra. How much of the Ti is bound with C in MAX phase and what fraction is in the oxide state. This is important because in many cases working with MAX phases and MXenes oxidation of the Ti becomes an issue.

Reply:

We appreciate the constructive suggestions from the reviewer and apologize for the negligence on our part. In response, we have re-conducted a series of XPS experiments, accompanied by a quantitative analysis to analyze the atomic bonding state of 1D-Ti₂AlC samples. The figure R1-R4 showed below illustrated the XPS spectra of 1D-Ti₂AlC synthesized at different temperature, along with the spectra of commercial Ti₂AlC powders for comparison. The corresponding quantitative analysis from XPS have been summarized into Table R1-R4, and we conducted a reevaluation of the results, and accordingly revised the manuscript.

According to these results, the Ti 2p region could be deconvoluted into the bonding of Ti-C and Ti-O respectively, matching the 2p_{3/2} peaks at binding energies of 454.3 and 458.1 eV, respectively, with their respective spin-orbit splitting 2p_{1/2} peaks that differing about 6 eV, where at 460.2, and 464.1 eV. The Ti-C fraction increased with higher synthesizing temperature, suggesting the increased purity. The Al 2p spectra shows Al-Ti and Al-O peaks at around 71.6 and 74.5 eV, respectively. The Al-O fraction is predominant over Al-Ti fraction, indicating that aluminum exhibits sensitivity to oxygen at elevated temperatures. The binding energies of C 1s at 281.3, 284.6, 286 and 288.6 eV represented C-Ti, C-C, C-O, and C=O, respectively. The O 1s results could be fitted with three curves to match O-Ti, O-Al and O-C bonds, peaking at around 529.9, 530.9 and 531.9 eV, respectively (ref. 65-67). The greater abundance of Al-O fraction is larger than Ti-O fraction signifies that Al is more prone to surface oxidation than titanium. The ratio of fraction related to the oxidation states were summarized in Table R5, consistent with the analysis above. In summary, the surface oxidation of both Ti and Al occurred during synthesis, with aluminum exhibiting a greater tendency for oxidation. Additionally, the Gibbs free energy of reactions for producing Al₂O₃ and TiO₂ is calculated in figure R5 to substantiate this observation.

Corresponding modifications reflecting the analysis above have been integrated into the relevant sections of the manuscript.

Figure R1. Deconvolution of (a)Ti 2p, (b) Al 2p and (c) O 2p XPS spectra for 1D-Ti₂AlC samples at reaction temperature of 800 °C.

Figure R2. Deconvolution of (a)Ti 2p, (b) Al 2p and (c) O 2p XPS spectra for 1D-Ti₂AlC samples at reaction temperature of 850 °C.

Figure R3. Deconvolution of (a)Ti 2p, (b) Al 2p and (c) O 2p XPS spectra for 1D-Ti₂AlC samples at reaction temperature of 900 °C.

Figure R4. Deconvolution of (a) Ti 2p, (b) Al 2p and (c) O 2p XPS spectra for commercial Ti_2AlC powders.

Figure R5. The Gibbs free energy of reaction R1 and R2 calculated using Factsage software.

In fact, with regard to the molten salt synthesis method we employed during the high temperature synthesis of MAX phases, the oxidation of reactants is not significantly severe. As supported by ref. 13, 18, 19, and other pertinent studies, molten salts melt at temperatures lower than those required for the synthesis of MAX phases, thereby encapsulating the reactants and isolating them from atmospheric oxygen. Moreover, we provided a continuous vacuum environment during the reaction to minimize the exposure

of reactants to atmospheric oxygen to the greatest extent possible.

Reference.

- 13.** Shao, H. *et al.* Synthesis of MAX Phase Nanofibers and Nanoflakes and the Resulting MXenes. *Adv. Sci.* 10, doi:10.1002/advs.202205509 (2022).
- 18.** Li, S. *et al.* Advances in Molten Salt Synthesis of Non-oxide Materials. *Energy Environ. Mater.* 6, doi:10.1002/eem2.12339 (2022).
- 19.** Dash, A., Vaßen, R., Guillon, O. & Gonzalez-Julian, J. Molten salt shielded synthesis of oxidation prone materials in air. *Nat. Mater.* 18, 465-470, doi:10.1038/s41563-019-0328-1 (2019).
- 65.** Zhang, Z. *et al.* Probing the oxidation behavior of Ti₂AlC MAX phase powders between 200 and 1000 °C. *JOURNAL OF THE EUROPEAN CERAMIC SOCIETY* 37, 43-51, doi: 10.1016/j.jeurceramsoc.2016.08.004 (2017).
- 66.** Magnuson, M. *et al.* Electronic structure and chemical bonding in Ti₂AlC investigated by soft x-ray emission spectroscopy. *PHYSICAL REVIEW B* 74, doi:10.1103/PhysRevB.74.195108 (2006).
- 67.** Naslund, L.-A., Persson, P. O. A. & Rosen, J. X-ray Photoelectron Spectroscopy of Ti₃AlC₂, Ti₃C₂T_z, and TiC Provides Evidence for the Electrostatic Interaction between Laminated Layers in MAX-Phase Materials. *JOURNAL OF PHYSICAL CHEMISTRY C* 124, 27732-27742, doi:10.1021/acs.jpcc.0c07413 (2020).

Table R1. Parameters obtained for the Ti 2p curve fitting of 1D-Ti₂AlC samples at reaction temperature of 800 °C.

Region	BE (eV)	FWHM (eV)	Fraction	Assigned to
Ti 2p_{3/2}(2p_{1/2})	454.2(460.7)	2.4(0.8)	22.7	Ti-C
	458.9(464.0)	2.8(3.1)	77.3	Ti-O
C 1s	281.3	2.2	10.9	Ti-C
	284.6	1.7	49.7	C-C
	286	1.5	21.2	C-O
	288.3	3.3	18.2	C=O
Al 2p_{3/2}(2p_{1/2})	71.0(71.6)	0.5(0.9)	14.2	Al-Ti
	73.9(74.5)	2.5(1.9)	85.8	Al-O
O 1s	529.5	1.4	22.3	Ti-O
	530.6	1.8	27.7	Al-O
	532	2.9	49.9	C-O

Table R2. Parameters obtained for the Ti 2p curve fitting of 1D-Ti₂AlC samples at reaction temperature of 850 °C.

Region	BE (eV)	FWHM (eV)	Fraction	Assigned to
Ti 2p_{3/2}(2p_{1/2})	454.4(460.7)	2.3(1.7)	30.8	Ti-C
	458.1(463.7)	2.3(2.5)	69.2	Ti-O
C 1s	281.4	0.8	3.8	Ti-C
	284.8	2.1	85.3	C-C
	286.5	0.9	6.1	C-O
	288.7	1.1	4.9	C=O
Al 2p_{3/2}(2p_{1/2})	71.7(72.4)	0.9(3.4)	9.1	Al-Ti
	74.1(74.8)	2.1(1.7)	90.1	Al-O
O 1s	529.7	1.5	16.8	Ti-O
	530.8	3.4	44.3	Al-O
	531.1	2.1	38.9	C-O

Table R3. Parameters obtained for the Ti 2p curve fitting of 1D-Ti₂AlC samples at reaction temperature of 900 °C.

Region	BE (eV)	FWHM (eV)	Fraction	Assigned to
Ti 2p_{3/2}(2p_{1/2})	454.4(460.2)	2.2(2)	30.9	Ti-C
	458.1(463.8)	2.7(2.5)	69.1	Ti-O
C 1s	281.3	1.9	6.1	Ti-C
	284.6	2	67	C-C
	285.7	1.9	16.3	C-O
	287.6	3.2	10.6	C=O
Al 2p_{3/2}(2p_{1/2})	71.6(72.2)	3.4(0.7)	6.1	Al-Ti
	74.1(74.6)	2.1(1.2)	93.9	Al-O
O 1s	529.9	1.6	15.7	Ti-O
	530.8	3.2	55.5	Al-O
	531.1	1.7	28.8	C-O

Table R4. Parameters obtained for the Ti 2p curve fitting of commercial Ti₂AlC powders.

Region	BE (eV)	FWHM (eV)	Fraction	Assigned to
Ti 2p_{3/2}(2p_{1/2})	454.1(459.7)	1.9(3.4)	58.7	Ti-C
	458.3(464.1)	1.1(2.4)	41.3	Ti-O
C 1s	281.6	2	10.9	Ti-C
	284.9	1.5	59.7	C-C
	286	1.4	13.5	C-O
	288.7	2.4	15.9	C=O
Al 2p_{3/2}(2p_{1/2})	70.6(71.2)	0.9(0.5)	13.3	Al-Ti
	73.7(74.3)	1.8(3)	86.7	Al-O
O 1s	530.1	0.7	2.2	Ti-O
	530.9	1.2	20.8	Al-O
	532.5	2.8	77	C-O

Table R5. Summary of comparisons of oxidation state from XPS results.

Ratio	800 °C	850 °C	900 °C	Standard
Ti-O: Ti-C	3.4	2.25	2.23	0.7
Al-O: Al-Ti	6	9.9	15.4	15.3
O-Al: O-Ti	1.24	2.6	3.5	9.45

In general, I would like to see many more SAD diffraction patterns. For example I would like to see SAD for Fig. S18. The same for Figs. 1f and 1g.

Reply:

Thank you for your suggestion. In response to your suggestion, we supplemented the SAED data using the same sample as Figures 1f and 1g, specifically conducting selected area electron diffraction (SAED) photography in Figure R6. However, due to the abundance of samples on the copper grid, locating the exact positions corresponding to Figures 1f and 1g was unfeasible. Consequently, we applied Fourier transforms to Figures 1f and 1g, presenting the results in Figure R7. Similarly, a Fourier transform was applied to Figure S18 to obtain the electron diffraction pattern. The corresponding FFT images of 1D-Ti₃SiC₂ and 1D-Ti₂SnC have also incorporated into the Supplementary Information.

Figure R6. SAED patterns of 1D-Ti₂AlC with incident beam along the [1 $\bar{1}$ 20] direction.

Figure R7. (a) TEM image of 1D-Ti₂AlC from figure 1f in manuscript, and (b) is its corresponding FFT pattern; (c) TEM image of 1D-Ti₂AlC from figure 1g in manuscript, and (d) is its corresponding FFT pattern.

Figure R8. (a) TEM image of 1D-Ti₃AlC₂ from figure S18 in manuscript, and (b) is its corresponding FTT pattern; (c) TEM image of 1D-Ti₃AlC₂ from another area in 1D-Ti₃AlC₂, and (d) is its corresponding FTT pattern.

Figure R9. (a) TEM image of 1D-Ti₃SiC₂, and (b) is its corresponding FTT pattern.

Figure R10. (a) TEM image of 1D-Ti₂SnC, and (b) is its corresponding FTT pattern.

In Fig. 2f the authors claim the large exothermic peak is due to the evaporation of the salt. Normally evaporation is endothermic. What am I missing?

Reply:

We regret to acknowledge inaccuracies in the analysis presented within the content of our manuscript, specifically regarding the characterization of molten salt evaporation as an endothermic process. Accordingly, we have undertaken a thorough reevaluation of our findings, including the re-conduction of TG-DSC tests, followed by a comprehensive reanalysis of the data obtained. The corrections have been modified in both the content and figures (Figure R11) as follows.

Differential scanning calorimetry (DSC) and thermogravimetric (TG) performed on reactants in Ar atmosphere to corroborate the formation process of 1D-Ti₂AlC (Fig. 2e). There were several stages during the reaction based on the DSC curve. The sharp endothermic peak emerged around 656 °C indicated the melt of the salt mixture and Al atoms. The third stage represented the formation of Ti-Al intermetallic compounds. The exothermic peak observed at 871 °C in the next stage accorded well with the continuous formation of Ti₂AlC along with the XRD results above. The endothermic peak above 1018 °C represented the vaporization of molten salts, and the caused mass loss led to the absence of NaCl and KCl peaks in the XRD results (ref. 33, 35-37).

For synthesizing other 1D-MAX phases, the major difference is the position of exothermic peak, representing their corresponding formation temperature. Noting that the peaks in DSC analysis tend to shift to higher temperatures due to the elevated heating rate, which can introduce a slight discrepancy between DSC and XRD results. However, these deviations are within an acceptable range for such thermal analyses.

Reference.

33. Nadimi, H., Soltanieh, M. & Sarpoolaky, H. Molten salt shielded synthesis and formation mechanism of Ti₂AlC in NaCl–KCl medium. *Ceram. Int.* **48**, 9024-9029, doi: 10.1016/j.ceramint.2021.12.084 (2022).
35. Shuck, C. E. *et al.* Effect of Ti₃AlC₂ MAX Phase on Structure and Properties of Resultant Ti₃C₂T_x MXene. *ACS APPLIED NANO MATERIALS* **2**, 3368-3376,

doi:10.1021/acsnm.9b00286 (2019).

36. Etebarian, S., Sarpoolaky, H., Rezaie, H. R. & Velashjerdi, M. Synthesis of Ti_3SiC_2 MAX phase powder through molten salt method. *INTERNATIONAL JOURNAL OF APPLIED CERAMIC TECHNOLOGY* **20**, 2166-2174, doi:10.1111/ijac.14358 (2023).

37. Kang, Y. J., Fey, T. & Greil, P. Synthesis of Ti_2SnC MAX Phase by Mechanical Activation and Melt Infiltration. *Adv. Eng. Mater.* **14**, 85-91, doi:10.1002/adem.201100186 (2012). Li, S. *et al.* Advances in Molten Salt Synthesis of Non-oxide Materials. *Energy Environ. Mater.* **6**, doi:10.1002/eem2.12339 (2022).

Figure R11. DSC and TG curve of a mixture of (a) Ti, Al powder and TiC nanofibers, (b) Al powder and TiC nanofibers, (c) Ti, Si powder and TiC nanofibers, and (d) Ti, Sn powder and TiC nanofibers in the presence of the KCl–NaCl eutectic salt under the protection of Ar gas at a rate of 10 °C/min.

The claim that the composites could be used in many applications in where low friction was important is very debatable. Which surface do the authors propose to use their composites? The large usable area is the top area with is pure Cu, no?

Reply:

We greatly appreciate your valuable suggestions. In this study, the CALF composite generated from copper and 1D- Ti_2AlC were designed to enhance the mechanical and physiochemical performance of copper, who extensively utilized across various areas but with inherent limitations. As discussed in Section 4, employing 1D- Ti_2AlC as an enhancement phase compensates for copper's deficiencies (ref. 39-41, 45, 49).

Experimental and literature reviews indicated that augmenting 1D-Ti₂AlC improves copper's wear resistance (ref. 50-52). Initially, our focus was on CALF composite's cross-sectional wear resistance, now we complemented with surface wear resistance assessments in figure R12 according to your suggestions. The results indicated superior anti-friction characteristics in both surface and cross-section of CALF compared to pure copper, suggesting its potential for practical applications.

Figure R12. Coefficient of friction curves conducted from (a) surface of samples, and (b) cross-section of samples along different directions.

Moreover, the large usable area is not pure copper, where is previously ambiguous in our manuscript. Investigations reveal that copper would react with MAX phases by interdiffusion reactions, with Al from the MAX phase diffusing into copper, resulting in the formation of a Cu (Al) solid solution layer and a TiC layer as shown in equation 3. In our cases, the CALF is composed of Cu (Al) film layer and TiC nanofiber matrix layer, as illustrated in figure 4 (b) and 4 (c). The XRD results in Figure R13 illustrates the phases of both cross-section and surface of CALF composites, which are consistent with the discussion above. For the Cu (Al) solid solution in the top area of CALF, the wear mechanism transitions from abrasive and adhesive wear to mild adhesive wear, achieving lower friction coefficients. Additionally, particle reinforcement and minor solubilization of Ti₂AlC particles in the copper matrix elevate the friction material's hardness, enhancing wear resistance (ref. 38, 58, 59).

Reference.

38. Jahani, A., Jamshidi Aval, H., Rajabi, M. & Jamaati, R. Effects of Ti₂SnC MAX Phase on Microstructure, Mechanical, Electrical, and Wear Properties of Stir-Extruded Copper Matrix Composite. *Adv. Eng. Mater.* 25, doi:10.1002/adem.202201463 (2023).
39. Li, M. *et al.* Tensile behavior and strengthening mechanism in ultrafine TiC_{0.5} particle reinforced Cu–Al matrix composites. *J. Alloys Compd.* **628**, 186-194, doi: 10.1016/j.jallcom.2014.10.123 (2015).
40. Si, P. *et al.* Tailorable Metal–Ceramic (Cu–TiC_{0.5}) Layered Electrode with High

Mechanical Property and Conductivity. *ACS Appl. Mater. Interfaces* **11**, 44413-44420, doi:10.1021/acsami.9b13219 (2019).

41. Zhang, X. *et al.* Investigations on the interface-dominated deformation mechanisms of two-dimensional MAX-phase $\text{Ti}_3\text{Al}(\text{Cu})\text{C}_2$ nanoflakes reinforced copper matrix composites. *Acta Mater.* **240**, doi: 10.1016/j.actamat.2022.118363 (2022).

45. Yang, M., Weng, L., Zhu, H., Fan, T. & Zhang, D. Simultaneously enhancing the strength, ductility and conductivity of copper matrix composites with graphene nanoribbons.

49. *Carbon* **118**, 250-260, doi: 10.1016/j.carbon.2017.03.055 (2017).

Yang, Z. *et al.* Electrical conductivities and mechanical properties of Ti_3SiC_2 reinforced Cu-based composites prepared by cold spray. *J. Alloys Compd.* **946**, doi: 10.1016/j.jallcom.2023.169473 (2023).

50. Yang, Z. *et al.* Improving the mechanical properties and electrical conductivity of cold-sprayed Cu- Ti_3SiC_2 composite by friction stir processing. *Compos Part A-Appl S* **173**, doi: 10.1016/j.compositesa.2023.107698 (2023).

51. Salvo, C. *et al.* Microstructure, electrical and mechanical properties of Ti_2AlN MAX phase reinforced copper matrix composites processed by hot pressing. *Mater. Charact.* **171**, doi: 10.1016/j.matchar.2020.110812 (2021).

52. Yang, D., Zhou, Y., Yan, X., Wang, H. & Zhou, X. Highly conductive wear resistant Cu/ $\text{Ti}_3\text{SiC}_2(\text{TiC}/\text{SiC})$ co-continuous composites via vacuum infiltration process. *J. Adv. Ceram.* **9**, 83-93, doi:10.1007/s40145-019-0350-4 (2020).

58. Wang, F. *et al.* Influence of two-step ball-milling condition on electrical and mechanical properties of TiC-dispersion-strengthened Cu alloys. *Mater. Des.* **64**, 441-449, doi: 10.1016/j.matdes.2014.08.027 (2014).

59. Xu, G. *et al.* Continuous electrodeposition for lightweight, highly conducting and strong carbon nanotube-copper composite fibers. *Nanoscale* **3**, 4215-4219, doi:10.1039/c1nr10571j (2011).

Figure R13. XRD patterns of (a) surface, and (b) cross-section of CALF composites.

Other problems

The references do not refer to any of the pioneering work by Barsoum on the MAX phases. Also in certain places the authors make claims with no refs. For example, in Fig. S4 and

S10e they claim that Raman spectrum agrees with previous work. What previous work?
Pls reference.

Reply:

Thanks for your valuable comments. We express our profound gratitude for your invaluable suggestions. Indeed, Professor Barsoum and his coworkers are recognized pioneers in the research field of MAX phases. In fact, the conceptual framework and the hypotheses of our manuscript were significantly inspired by Professor Barsoum's seminal contributions, ranging from the synthesis of three-dimensional MAX phases to the derivation of two-dimensional MXenes. This inspiration led us to speculate on the feasibility of fabricating one-dimensional MAX phases. Admittedly, our citation of Professor Barsoum's work was limited in scope, encompassing only a few references such as ref. 9 and 29. Consequently, we have augmented our manuscript with additional references to Professor Barsoum's other works to increase the credibility of certain statements, including ref. 8-12, 27, 67-69.

In accordance with your recommendations, we have carefully reviewed the whole manuscript and strategically integrated additional references at some statements. We hope this endeavor could make arguments more precise.

We apologize for the imprecise expressions previously articulated within the captions of those figures. Specifically, the calculated positions of the Raman peaks depicted in Figure S4 were derived from, and should have been correctly attributed to the work of Professor Barsoum as referenced in ref. 69. Furthermore, the commentary associated with Figure S10e was intended to convey a comparison with literature reports from other researchers, rather than our own prior publications. Amendments have been made to rectify these descriptions, and the appropriate references have now been duly incorporated to ensure accurate attribution and context.

Reference.

8. Ding, H. *et al.* Chemical scissor-mediated structural editing of layered transition metal carbides. *Science* 379, 1130-1135, doi:10.1126/science.add5901 (2023).
9. Sokol, M., Natu, V., Kota, S. & Barsoum, M. W. On the Chemical Diversity of the MAX Phases. *TRENDS IN CHEMISTRY* 1, 210-223, doi:10.1016/j.trechm.2019.02.016 (2019).
10. Tzenov, N. V. & Barsoum, M. W. Synthesis and characterization of Ti_3AlC_2 *JOURNAL OF THE AMERICAN CERAMIC SOCIETY* 83, 1551-1551 (2000).
11. Naguib, M. *et al.* Two-Dimensional Nanocrystals Produced by Exfoliation of Ti_3AlC_2 . *Adv. Mater.* 23, 4248-4253, doi:10.1002/adma.201102306 (2011).
12. Lukatskaya, M. R. *et al.* Cation Intercalation and High Volumetric Capacitance of Two-Dimensional Titanium Carbide. *Science* 341, 1502-1505, doi:10.1126/science.1241488 (2013).
27. Barsoum, M. W., Ali, M. & El-Raghy, T. Processing and characterization of Ti_2AlC , Ti_2AlN , and $Ti_2AlC_{0.5}N_{0.5}$. *Metall. Mater. Trans. A* 31, 1857-1865, doi:10.1007/s11661-006-0243-3 (2000).
67. Naslund, L.-A., Persson, P. O. A. & Rosen, J. X-ray Photoelectron Spectroscopy of Ti_3AlC_2 , $Ti_3C_2T_z$, and TiC Provides Evidence for the Electrostatic Interaction between Laminated Layers in MAX-Phase Materials. *JOURNAL OF PHYSICAL CHEMISTRY C* 124,

27732-27742, doi: 10.1021/acs.jpcc.0c07413 (2020).

68. Barsoum, M. W. & El-Raghy, T. Synthesis and Characterization of a Remarkable Ceramic: Ti_3SiC_2 . *J. Am. Ceram. Soc.* **79**, 1953-1956, doi:10.1111/j.1151-2916.1996.tb08018.x (1996).

69. Leaffer, O. D., Gupta, S., Barsoum, M. W. & Spanier, J. E. On Raman scattering from selected M_2AC compounds. *JOURNAL OF MATERIALS RESEARCH* **22**, 2651-2654, doi:10.1557/JMR.2007.0376 (2007).

Reference for conductivity of pure Cu?

Reply:

In fact, the electrical conductivity of pure copper reported in this study was directly measured by four-probe resistance measurement (RTS-9, Guangzhou Four Probe Technology, China), and the relevant testing procedures detailed within the experimental section. The specific type of copper utilized is also described in the experimental part.

Furthermore, we have supplemented our report with additional references concerning the electrical conductivity of pure copper, such as ref.46 and 47.

Reference.

46. Zhang, X. *et al.* Achieving high strength and high ductility in metal matrix composites reinforced with a discontinuous three-dimensional graphene-like network. *Nanoscale* **9**, 11929-11938, doi:10.1039/c6nr07335b (2017).

47. Lu, L., Shen, Y. F., Chen, X. H., Qian, L. H. & Lu, K. Ultrahigh strength and high electrical conductivity in copper. *SCIENCE* **304**, 422-426, doi:10.1126/science.1092905 (2004). Ding, H. *et al.* Chemical scissor-mediated structural editing of layered transition metal carbides. *Science* **379**, 1130-1135, doi:10.1126/science.add5901 (2023).

Reply to Reviewer 2

Yuting Li et al. provided lateral thinking to fabricate 1D MAX nanofiber, and the research has innovativeness and systematicness. Then, we concerned several questions as follows:
1. Why authors choose to fabricate TiC nanofibers first but not to use commercial carbon nanofibers? Is the reason the existence of amorphous carbon in carbon nanofibers?

Reply:

We greatly appreciate your affirmation of our work and commit to addressing each of your inquiries with thorough and considered responses.

Firstly, the purpose of this work is to provide a viable strategy for synthesizing 1D-MAX phases, which is the conformal synthesis. As we mentioned in the last paragraph of manuscript, each step of the conformal synthesis is flexible and adaptable, including the selection of precursor templates, the choice of reactants, and the selection of synthesis methods, among others. Indeed, in the decades following the discovery of MAX phases, various combinations have been utilized as reactants for synthesizing Ti_2AlC , such as $Ti/Al_4C_3/C$, $Ti/Al/C$, $Ti/Al/C/TiC$, and $Ti/C/TiAl$ (ref. 10, 27, 30).

Then, addressing your concerns, both C and TiC can be used as reactants in the conventional methods of synthesizing MAX phases. For instance, references 13, 17 employed carbon as the carbon source, while references 16 chose TiC as the carbon source. Regarding our study, there are two reasons that we firstly chose TiC nanofibers. On the one hand, the crystal structure of MAX phase crystal structure consists of alternatively stacked 'MX' layers and 'A' layers. The well-known 2D-material, MXene, is obtained by exfoliating the A layers from bulk MAX phase. Consequently, by employing reverse thinking, we considered the feasibility of intercalating the 'A' layer into the one-dimensional form of MX to re-generate 1D-MAX phases. On the other hand, numerous pioneering studies have demonstrated that TiC frequently serves as the most stable intermediate phase in the Ti-Al-C MAX phase system (ref. 25, 27, 28, 30-33). The last step of the formation of MAX phase through the reaction between TiC and TiAl compounds, macroscopically. This synthesis dynamic entails that upon employing Ti, Al, and C as reactants, a preferential reaction pathway leads to the initial formation of TiC, while a reaction between Ti and Al to yield TiAl is a competitive pathway. This competitive dynamic is illustrated by the calculated Gibbs free energy in Figure R2. Therefore, theoretically, TiC nanofibers as reactant reduces those side reactions compared to using CNF, thus leading to a superior purity of MAX phases, consequently.

Additionally, we conducted conformal synthesis under similar conditions (Table R1) using CNF for comparative analysis. As illustrated in Figure R2, due to the propensity of Ti metal to react with carbon, it is postulated that a TiC layer formed on the surface of the CNF. Subsequent reactions between TiC and Ti-Al metallic compounds led to the loss of the long-range ordered morphology of the fibers, particularly when an excess of metal was present, severely impacting their surface morphology. This is a situation that the use of TiC could avoid. Moreover, XRD analysis revealed a lower yield of the MAX phase, indicating that the use of CNF may necessitate higher temperatures or longer reaction times for comparable products formation.

Lastly, CNF, serving as the carbon source during the synthesis of MAX phases, primarily contributes to the formation of TiC as an intermediate product. Actually, the amorphous part in CNF would not influence the reaction. Other studies have also demonstrated that various forms of carbon (such as CNTs, graphene, and carbon black) are capable of reacting with Ti to form TiC (ref. 49). We hope this clarifies your concerns.

Reference.

10. Tzenov, N. V. & Barsoum, M. W. Synthesis and characterization of Ti_3AlC_2 *JOURNAL OF THE AMERICAN CERAMIC SOCIETY* 83, 1551-1551 (2000).
13. Shao, H. *et al.* Synthesis of MAX Phase Nanofibers and Nanoflakes and the Resulting

MXenes. *Adv. Sci.* **10**, doi:10.1002/advs.202205509 (2022).

17. Liu, Y. *et al.* Facile synthesis of hollow Ti_3AlC_2 microrods in molten salts via Kirkendall effect. *J. Adv. Ceram.* **11**, 1491-1497, doi:10.1007/s40145-022-0616-0 (2022).

25. Lee, H., Kim, S. Y., Lee, Y. I. & Byun, J. Synthesis and reaction path of Ti-Al-C MAX phases by reaction with Ti-Al intermetallic compounds and TiC. *J. Am. Ceram. Soc.*, doi:10.1111/jace.19217 (2023).

27. Barsoum, M. W., Ali, M. & El-Raghy, T. Processing and characterization of Ti_2AlC , Ti_2AlN , and $\text{Ti}_2\text{AlC}_{0.5}\text{N}_{0.5}$. *Metall. Mater. Trans. A* **31**, 1857-1865, doi:10.1007/s11661-006-0243-3 (2000).

28. Liu, Z., Xu, J., Xi, X., Luo, W. & Zhou, J. Molten salt dynamic sealing synthesis of MAX phases (Ti_3AlC_2 , Ti_3SiC_2 et al.) powder in air. *Ceram. Int.* **49**, 168-178, doi: 10.1016/j.ceramint.2022.08.325 (2023).

30. Xiao, Z. *et al.* Investigation of Ti_2AlC formation mechanism through carbon and TiAl diffusional reaction. *J. Eur. Ceram. Soc.* **38**, 1246-1252, doi: 10.1016/j.jeurceramsoc.2017.10.039 (2018).

31. Yang, H. *et al.* A new insight into heterogeneous nucleation mechanism of Al by non-stoichiometric TiC_x . *Acta Mater.* **233**, doi: 10.1016/j.actamat.2022.117977 (2022).

32. Wang, Z. *et al.* Investigation on the in-situ reaction mechanism of $\text{Ti}_2\text{AlC}/\text{TiAl}$ composite prepared by spark plasma sintering. *Mater. Charact.* **194**, doi: 10.1016/j.matchar.2022.112417 (2022).

33. Nadimi, H., Soltanieh, M. & Sarpoolaky, H. Molten salt shielded synthesis and formation mechanism of Ti_2AlC in NaCl-KCl medium. *Ceram. Int.* **48**, 9024-9029, doi: 10.1016/j.ceramint.2021.12.084 (2022).

49. Thomas, T. & Bowen, C. R. Effect of particle size on the formation of Ti_2AlC using combustion synthesis. *Ceram. Int.* **42**, 4150-4157, doi: 10.1016/j.ceramint.2015.11.088 (2016).

Figure R1. The Gibbs free energies of 1 mol TiC and 1 mol TiAl produced from reactants calculated by Factsage software.

Figure R2. Conformal synthesis using CNF as carbon source. The surface morphologies of (a) CNF- Ti_2AlC , and (b) CNF- Ti_3SiC_2 synthesized as the procedure displayed in Table R1. The XRD patterns of (c) CNF- Ti_2AlC , and (d) CNF- Ti_3SiC_2 obtained with different synthesizing temperature.

Table R1.

Preparation of 1D-MAX phases from TiC and CNF nanofibers template in molten salts environment

MAX Phase	Carbon source	Ratio of adding atoms (by mol)	Ratio of atoms and eutectic salts (by mol)	T (°C)	Reaction time (hours)
Ti_2AlC	CNF	Ti: Al = 2:1.2	1: 4	900	2
	TiC NF	Ti: Al = 1:1.2			
Ti_3SiC_2	CNF	Ti: Si: Al = 2.5: 1.2: 0.1	1: 10	1100	1
	TiC NF	Ti: Si: Al = 1: 1.2: 0.1			

2. What are the impurities in MAX nanofibers? If the impurities include ceramic phases like TiC, how did you eliminate them using HCl?

Reply:

As mentioned in part 2 of the manuscript, the primary impurities during the synthesis of MAX phases are predominantly intermetallic compounds, such as TiAl , TiAl_3 , Ti_3Al , along with minor quantities of Al_2O_3 and TiC, as showed in Figure S21, 26, and 33.

Indeed, the presence of TiC impurities is inevitable for all MAX phases synthesis,

which is not likely to eliminate completely (ref.10, 13, 27, 68). Our efforts thus directed towards maximizing the purity of the MAX phase products. For conformal synthesis in this work, one of the merits is that the amounts of metal reactants used is excessive, since the impurities from them could be easily washed after synthesis. Therefore, the TiC nanofiber is capable of undergoing a maximal transformation into MAX phases with the excess of other reactants, consequentially increasing the purity of the resultant product. Additionally, increasing reaction time and providing a consistent vacuum environment during synthesis are the other ways to avoid the presence of other impurities by facilitating the reaction.

The application of HCl is primarily aimed at removing intermetallic compounds, e.g., TiAl, Ti₃Al, and TiAl₃, as they readily react with the acid and dissolve in the solution. While the products MAX phases, which exhibit excellent acid resistance are thereby preserved (ref. 8, 9, 10). This exemplifies one of the inherent advantages of the conformal synthesis.

Reference.

8. Ding, H. *et al.* Chemical scissor-mediated structural editing of layered transition metal carbides. *Science* 379, 1130-1135, doi:10.1126/science.add5901 (2023).
9. Sokol, M., Natu, V., Kota, S. & Barsoum, M. W. On the Chemical Diversity of the MAX Phases. *TRENDS IN CHEMISTRY* 1, 210-223, doi: 10.1016/j.trechm.2019.02.016 (2019).
10. Tzenov, N. V. & Barsoum, M. W. Synthesis and characterization of Ti₃AlC₂ *JOURNAL OF THE AMERICAN CERAMIC SOCIETY* 83, 1551-1551 (2000).
13. Shao, H. *et al.* Synthesis of MAX Phase Nanofibers and Nanoflakes and the Resulting MXenes. *Adv. Sci.* 10, doi:10.1002/advs.202205509 (2022).
27. Barsoum, M. W., Ali, M. & El-Raghy, T. Processing and characterization of Ti₂AlC, Ti₂AlN, and Ti₂AlC_{0.5}N_{0.5}. *Metall. Mater. Trans. A* 31, 1857-1865, doi:10.1007/s11661-006-0243-3 (2000).
68. Barsoum, M. W. & El-Raghy, T. Synthesis and Characterization of a Remarkable Ceramic: Ti₃SiC₂. *J. Am. Ceram. Soc.* 79, 1953-1956, doi:10.1111/j.1151-2916.1996.tb08018.x (1996).

3. What length do the MAX nanofibers extend? And the length-to-diameter of the MAX nanofibers should be given in the text or graph sections.

Reply:

We are grateful for your valuable suggestions.

Electrospinning is one of the common methods for obtaining nanofibers driven by electrical forces. During the electrospinning process, a high voltage is applied at the tip of the nozzle, the droplet at the tip stretches into a cone shape (known as the Taylor cone) by increased voltage. Once the electrostatic force is sufficient to overcome the surface tension of the droplet, a fine charged jet is ejected from the Taylor cone and undergoes a series of unstable motions in the air, rapidly forming fine fibers as the solvent rapidly evaporates. In this process, nanofibers are formed and typically deposited over a large area on the collection part, thereby directly forming a two-dimensional membrane. Consequently, the extent to which a single fiber can extend is generally not a concern with this method. Actually, drawspinning is suitable for producing ultra-long single nanofibers or nanofiber nets, although its productivity is much lower than electrospinning (ref. 5, 60, 61).

The length of MAX phase nanofibers thus is dependent to the length of the as-spun nanofibers from electrospinning. Indeed, it proves challenging to ascertain the specific extent to which an individual fiber can elongate, since the as-spun mats are typically disordered unless specific measures and effective techniques are employed to mass-produce aligned nanofibers with sufficient fiber length (ref. 5, 60, 61). We endeavored to identify a segment of the MAX phase nanofiber membrane for tracking a single fiber, as illustrated in the figure R3 below. It could be seen that the obtained 1D-MAX phase could at least extent to the millimeter-level, and even longer to the meter-level length with larger collector. The reasons we chose electrospinning to produce 1D-template are its high productivity and simplicity of post-processing.

The radius of the 1D-MAX was assessed by figure R4 as a reference for our statistical analysis, as showed below. The average diameter of 1D-MAX is about 150nm, which is a typical diameter obtained by electrospinning.

Reference.

5. Huang, Y., Song, J., Yang, C., Long, Y. & Wu, H. Scalable manufacturing and applications of nanofibers. *Mater. Today* **28**, 98-113, doi: 10.1016/j.mattod.2019.04.018 (2019).

60. Zhang, F., Si, Y., Yu, J. & Ding, B. Electrospun porous engineered nanofiber materials: A versatile medium for energy and environmental applications. *Chem. Eng. J.* **456**, doi: 10.1016/j.cej.2022.140989 (2023).

61. Shi, S. *et al.* Recent Progress in Protective Membranes Fabricated via Electrospinning: Advanced Materials, Biomimetic Structures, and Functional Applications. *Adv. Mater.* **34**, doi:10.1002/adma.202107938 (2022).

Figure R3. SEM images for tracing one single nanofiber, which is marked with orange dot line.

Figure R4. SEM images for measuring the average diameters of 1D-Ti₂AlC. (a) A long piece of Ti₂AlC nanofiber membrane attached on the SEM stage. The selected areas are enlarged in (b) to (e), and the corresponding surface morphologies are presented from (f) to (i). The statistics of (j) to (m) are analyzed from (f) to (i), respectively.

4. HCl can etch the MAX phase together, how did you avoid this problem?

Reply:

The answer to this question is actually mentioned in question 2. MAX phases exhibit negligible reactivity towards hydrochloric acid, sulfuric acid, nitric acid, and similar substances, owing to their commendable acid resistance (ref. 8-10). They typically react only with certain concentrations of hydrofluoric acid (HF) or within mixed solutions of lithium fluoride (LiF) and HCl, a process related to the preparation of MXenes (ref. 11-13). Consequently, the presence of hydrochloric acid alone does not affect the MAX phases.

Reference.

8. Ding, H. *et al.* Chemical scissor-mediated structural editing of layered transition metal carbides. *Science* 379, 1130-1135, doi:10.1126/science.add5901 (2023).
9. Sokol, M., Natu, V., Kota, S. & Barsoum, M. W. On the Chemical Diversity of the MAX Phases. *TRENDS IN CHEMISTRY* 1, 210-223, doi: 10.1016/j.trechm.2019.02.016 (2019).
10. Tzenov, N. V. & Barsoum, M. W. Synthesis and characterization of Ti₃AlC₂ *JOURNAL OF THE AMERICAN CERAMIC SOCIETY* 83, 1551-1551 (2000).
11. Naguib, M. *et al.* Two-Dimensional Nanocrystals Produced by Exfoliation of Ti₃AlC₂.

Adv. Mater. **23**, 4248-4253, doi:10.1002/adma.201102306 (2011).

12. Lukatskaya, M. R. *et al.* Cation Intercalation and High Volumetric Capacitance of Two-Dimensional Titanium Carbide. *Science* **341**, 1502-1505, doi:10.1126/science.1241488 (2013).

13. Shao, H. *et al.* Synthesis of MAX Phase Nanofibers and Nanoflakes and the Resulting MXenes. *Adv. Sci.* **10**, doi:10.1002/advs.202205509 (2022).

5. Fig. 4 d doesn't have the scaleplate, thus confusing readers by an unclear scale for the distance of element diffusion. What is important is what reaction happened between Ti₂AlC and Cu. The authors should provide the XRD result of 1-Ti₂AlC/Cu composite. Does anything else like Al₂O₃ exist in the composite because the Al in Ti₂AlC will be easy to combine with O during hot-pressing process.

Reply:

We greatly appreciate your valuable suggestions and apologize for our oversight. We have added the corresponding scale bar to Figure 4d and have conducted a thorough review of the entire document to prevent similar issues.

In this study, the CALF composite generated from copper and 1D-Ti₂AlC were designed to enhance the mechanical and physiochemical performance of copper, where 1D-Ti₂AlC served as reinforcement phase. Investigations reveal that copper would react with MAX phases by interdiffusion reactions, with Al from the MAX phase diffusing into copper, resulting in the formation of a Cu (Al) solid solution layer and a TiC layer as shown in equation R1 (ref. 39-41, 45, 49-52, 58, 59). In our cases, the CALF is composed of Cu (Al) film layer and TiC nanofiber matrix layer, as illustrated in figure 4 (b) and 4 (c). The XRD results in Figure R5 below illustrates the phases of both cross-section and surface of CALF composites. There are no Al₂O₃ and other oxides detected.

Since the whole process of hot-pressing is under vacuum environment and the reactants are closely packed in the reaction chamber, the oxidation of Al is barely existed. Besides, the oxidation properties of Ti₂AlC and other MAX phases were analyzed by other researches, which revealed that Al would combine O under high temperatures exposed in air (ref. 65). For our study, this concern could not be significantly worried.

Reference.

39. Li, M. *et al.* Tensile behavior and strengthening mechanism in ultrafine TiC_{0.5} particle reinforced Cu–Al matrix composites. *J. Alloys Compd.* **628**, 186-194, doi: 10.1016/j.jallcom.2014.10.123 (2015).

40. Si, P. *et al.* Tailorable Metal–Ceramic (Cu-TiC_{0.5}) Layered Electrode with High Mechanical Property and Conductivity. *ACS Appl. Mater. Interfaces* **11**, 44413-44420, doi:10.1021/acsami.9b13219 (2019).

41. Zhang, X. *et al.* Investigations on the interface-dominated deformation mechanisms of two-dimensional MAX-phase Ti₃Al(Cu)C₂ nanoflakes reinforced copper matrix composites. *Acta Mater.* **240**, doi: 10.1016/j.actamat.2022.118363 (2022).

45. Yang, M., Weng, L., Zhu, H., Fan, T. & Zhang, D. Simultaneously enhancing the strength, ductility and conductivity of copper matrix composites with graphene nanoribbons.

49. *Carbon* **118**, 250-260, doi: 10.1016/j.carbon.2017.03.055 (2017).
- Yang, Z. *et al.* Electrical conductivities and mechanical properties of Ti_3SiC_2 reinforced Cu-based composites prepared by cold spray. *J. Alloys Compd.* **946**, doi: 10.1016/j.jallcom.2023.169473 (2023).
50. Yang, Z. *et al.* Improving the mechanical properties and electrical conductivity of cold-sprayed Cu- Ti_3SiC_2 composite by friction stir processing. *Compos Part A-Appl S* **173**, doi: 10.1016/j.compositesa.2023.107698 (2023).
51. Salvo, C. *et al.* Microstructure, electrical and mechanical properties of Ti_2AlN MAX phase reinforced copper matrix composites processed by hot pressing. *Mater. Charact.* **171**, doi: 10.1016/j.matchar.2020.110812 (2021).
52. Yang, D., Zhou, Y., Yan, X., Wang, H. & Zhou, X. Highly conductive wear resistant Cu/ Ti_3SiC_2 (TiC/SiC) co-continuous composites via vacuum infiltration process. *J. Adv. Ceram.* **9**, 83-93, doi:10.1007/s40145-019-0350-4 (2020).
58. Wang, F. *et al.* Influence of two-step ball-milling condition on electrical and mechanical properties of TiC-dispersion-strengthened Cu alloys. *Mater. Des.* **64**, 441-449, doi: 10.1016/j.matdes.2014.08.027 (2014).
59. Xu, G. *et al.* Continuous electrodeposition for lightweight, highly conducting and strong carbon nanotube-copper composite fibers. *Nanoscale* **3**, 4215-4219, doi:10.1039/c1nr10571j (2011).
65. Zhang, Z. *et al.* Probing the oxidation behavior of Ti_2AlC MAX phase powders between 200 and 1000 °C. *JOURNAL OF THE EUROPEAN CERAMIC SOCIETY* **37**, 43-51, doi:10.1016/j.jeurceramsoc.2016.08.004 (2017)

Figure R5. XRD patterns of (a) surface, and (b) cross-section of CALF composites.

6. Why did not the Ti_2AlC react entirely to be TiC when composited with Cu at 1000 °C?

Reply:

Sorry for the ambiguous discussion in manuscript. Actually, it did react entirely with Cu to be TiC, which is a detail previously lacking in our manuscript.

The answer for question 5 could also be the explaining for this question, which is that copper would react with MAX phases by interdiffusion reactions. The composites are formed with Al from the MAX phase diffusing into copper, resulting in the formation of a Cu (Al) solid solution layer and a TiC layer as shown in equation R1. In our cases, the CALF

is composed of Cu (Al) film layer and TiC nanofiber matrix layer, as illustrated in figure 4 (b) and 4 (c). The XRD results in Figure R5 above illustrates the phases of both cross-section and surface of CALF composites.

REVIEWERS' COMMENTS:

Reviewer #1 (Remarks to the Author):

I have had a chance to look at the revised paper. Unfortunately, and as I expected, the authors have much more oxide than MAX. According to their Tables R1 to R3 the C atomic % are, respectively, 10, 4 and 6. Unless I am missing something obvious, I am not sure how this low C content would be consistent with a Ti₂AlC chemistry. As importantly, the fraction of Ti in the oxide state is significantly higher than the carbide state/phase. Again assuming I am not missing the obvious, the material the authors made is more of an oxide with a little MAX, than vice versa. I am not sure what the authors can do to rectify this problem, but a start would be to change the title to something like: TiO₂/Ti₂AlC composite + Cu, etc.

Reviewer #2 (Remarks to the Author):

The authors have addressed my questions, and now it can be accepted.

Reviewer #3 (Remarks to the Author):

Comments on the XPS results and their analysis:

The XPS analysis in the paper is inadequate and inaccurate, the vast majority of the cited references do not even include XPS data (Refs. 12, 15, 17, 19 and 66). The authors cite reference 67 which is the only reliable reference however they are not following the same calibration (using the Fermi level instead of the so-called adventitious carbon and not following the same fitting models with no justification to that). To reliably compare the surface oxidation of Ti and Al between the different samples, a careful analysis must be performed for the XPS data. I would advise the authors to follow the same peak fitting and calibration of Ref. 67 to obtain accurate and reliable data.

RESPONSE TO REVIEWERS' COMMENTS

Reviewer #1 (Remarks to the Author):

I have had a chance to look at the revised paper. Unfortunately, and as I expected, the authors have much more oxide than MAX. According to their Tables R1 to R3 the C atomic % are, respectively, 10, 4 and 6. Unless I am missing something obvious, I am not sure how this low C content would be consistent with a Ti₂AlC chemistry. As importantly, the fraction of Ti in the oxide state is significantly higher than the carbide state/phase. Again assuming I am not missing the obvious, the material the authors made is more of an oxide with a little MAX, than vice versa. I am not sure what the authors can do to rectify this problem, but a start would be to change the title to something like: TiO₂/Ti₂AlC composite + Cu, etc.

Thank you very much for your response. Although there appears to be a divergence between your perspective and what we intended to convey, we will provide a detailed elaboration in the hopes of aligning our understanding and gaining your acceptance.

Firstly, and most importantly, we are confident that the purity of the 1D-MAX phase synthesized in our study, which is not as stated by your comment, TiO₂/Ti₂AlC. Based on the XRD and refinement results, the content of Ti₂AlC in the product exceeds 90%, comparable to the purities reported in other literature on synthesized MAX phases (Fig. 1b, Table S2, and S3 in manuscript). Moreover, in literature related to the synthesis of MAX phases cited in manuscript and many other literature, including the pioneer works by Barsoum, XRD results are typically used to assess the purity of products, rather than relying on XPS analysis. Therefore, in terms of product purity, XRD results carry more authority than XPS results.

Figure. 3b, Rietveld refinement of the XRD pattern of 1D-Ti₂AlC MAX.

Table S2. Cell parameters and components of prepared 1D-MAX phases obtained by Rietveld analysis.

MAX Phase	Lattice parameter a (Å)	Lattice parameter c (Å)	wt. %	Reduced χ^2	Rp (%)	GOF	TiC wt. %
Ti ₂ AlC	3.0246	13.6486	95.2	2.83	9.722	1.68	4.8
Ti ₃ AlC ₂	3.0583	18.4233	93.7	3.63	11.596	2.15	6.3
Ti ₃ SiC ₂	3.0647	17.6866	90.5	3.33	8.592	1.83	9.5

Ti₂SnC	3.1508	13.8462	91.4	1.08	5.29	1.04	8.6
Table S3. Comparison of MAX phase products with various micro morphologies from other reports.							
Purity (%)	Micro morphology	Bulk morphology	Synthesis temperature (°C)	Method	Ref		
> 95	Short nanofibers	Powder	900	MS	13		
< 50	Short microfibrs	Powder	900	SHS	14		
< 50	Long microfibrs	Fiber membrane	900	MS	15		
> 90	Short microrods	Powder	1250	MS	17		
> 95	Long nanofibers	Nanofiber membrane	900	MS	This work		

Then, based on the quantitative analysis of the supplementary XPS data required by you, we acknowledge that the results indicate a higher oxygen content. However, this should not be interpreted as implying that the overall sample is predominantly composed of oxides. Here is the explanation. XPS is a surface analysis technique, and as directly cited from the *Handbook of X-ray Photoelectron Spectroscopy: A Reference Book of Standard Spectra for Identification and Interpretation of XPS Data*,

Because the mean free path of electrons in solids is very small, the detected electrons originate from only the top few atomic layers, making XPS a unique surface-sensitive technique for chemical analysis. Quantitative data can be obtained from peak heights or peak areas, and identification of chemical states often can be made from exact measurement of peak positions and separations, as well as from certain spectral features. (John F. Moulder, 1992)

The detection scale of XPS typically extends only a few atomic layers in thickness. Therefore, the quantitative analysis results from XPS only represent the surface of materials within a specific detection range and do not reflect the overall composition. In the context of our work, XPS analysis can only reveal the chemical state of the nanofiber surface, whereas XRD has a detection depth typically on the micron scale, which is more representative of the scale of the nanofiber membranes in this study, and thus provides a more accurate assessment as discussed above.

The subsequent discussion involves the XPS analysis of MAX phase materials in the related literature. Notably, XPS analysis is relatively scarce in studies focusing on synthesizing MAX phases, especially the quantitative XPS analysis, which is almost non-existent. Consequently, there may exist inaccuracies and errors in our analysis of our own XPS results, as mentioned by the third reviewer. Herein, we conducted additional review and synthesized the findings in the table provided below. Analysis of these literature sources reveals a consistent observation: in nearly all XPS spectra of MAX phases, both Ti and Al exhibit oxidized states. This phenomenon is commonly attributed to surface oxidation resulting from the exposure of MAX phase materials to laboratory air. In our study, the nanostructured one-dimensional materials, due to their large surface area, are more prone to adsorbing oxygen from the air, thus exhibiting higher levels of oxidized states in quantitative analysis. Once again, we would like to emphasize that XPS results could not directly determine

whether the bulk sample predominantly comprises oxides.

TableR1: The Binding energies of XPS results of MAX phases from different literature.

Region	Ref [1]	Ref [2]	Ref [3]	Ref [4]	Ref [5]	Ref [6]	Ref [7]	Ref [8]
Ti 2p	454.5		454-454.7	453.7				
Ti-C Ti 2p _{3/2} (2p _{1/2})		454.5(460.3)			452.2(460.3)	454(460.3)	454.75(460.75)	454.6
Ti-O Ti 2p _{3/2} (2p _{1/2})		459(464.6)			458(463.6)	458.6(464)	459(464.8)	459
C 1s	281			281.1-282				281.9
C-Ti		281.8	281		281.9	281.5	281.88	
C-C		284	284.9		284.8	285	285.2	
COH		287				286.7	286.5	
COO		289.3				289	289.8	
Al 2p	72.37		72		72.5			
Al-Ti Al 2p _{3/2} (2p _{1/2})		72.1		71.2		71.7	72.13(72.53)	72.5
Al-O Al 2p _{3/2} (2p _{1/2})		74.7		73.7(74.3)		74.0	74.4(74.8)	75.7

It is worth noting that in both the main text and the Supplementary Information, we have addressed the presence of unreacted species and potential impurities generated during the reaction process (Figure.2d, Figure S18; Figure S23, Figure S27 and Figure S30 for other 1D-MAX phases). Primarily, these include unreacted TiC and intermetallic compounds, while the minimal oxides produced during synthesis under vacuum conditions are predominantly Al₂O₃ rather than TiO₂. This is a common occurrence in MAX phase synthesis, and the thermodynamically preferred oxide is Al₂O₃. We have previously mentioned this in our point-by-point response. Therefore, it is expected that our XPS results show a significant proportion of oxidized states of aluminum, consistent with the literature discussed above. The purpose of using XPS analysis in our study was initially to indirectly supplement the XRD results by demonstrating that as the reaction temperature increases, the completeness of the reaction also improves, leading to purer products. Hence, the gradual increase in the C atomic% from R1 to R3 as shown can substantiate this point. In fact, we could also omit the use of XPS analysis in the main text, as it is not as objectively reliable for quantitative data compared to XRD.

Moreover, we acknowledge that there are indeed issues with our XPS analysis. Therefore, we directly used the products obtained at 900°C for re-testing XPS and sought expert analysis. Following the suggestions of the third reviewer and in response to your queries, we conducted an asymmetric fitting and quantitative analysis of the Ti-2p curve, the results of which are shown below. It is evident that when using asymmetric fitting, rather than the symmetric fitting we initially used, the content of Ti-C is significantly higher than that of Ti-O.

Additionally, to better address the amount of oxides present, we also added some SEM-EDS mapping of elements from the product, since the penetration depth of the electron beam could cover the whole fiber, including the full mapping of figure 1d in original manuscript and two new mapping using both silicon wafers and copper matrix as supporting substrates (Figure R2-R4). It is evident that the distribution of oxygen is significantly less than that of the other three elements from all the results. It is worth noting that the carbon element from the carbon cloth in the copper matrix substrate and the oxygen element from the silicon wafer may be detected by EDS, which means that

the quantitative analysis ratios obtained may not be entirely accurate and should be considered as reference data only.

Thus, we believe that the original data of single nanofiber, such as some TEM and HRTEM results in the manuscript, are used to verify whether the nanofibers are MAX phase from a microscopic perspective. In contrast, XRD and TG-DTA data are used to confirm the product from a macroscopic perspective. Combining these results with these new results could help us prove the successful preparation of 1D-MAX phases.

Figure R1. Deconvolution of Ti 2p XPS spectra for 1D-Ti₂AlC samples at reaction temperature of 900 °C.

Table R2. The parameters obtained from Ti-2p curve fitting.

Region	BE (eV)	FWHM (eV)	Fraction	Assigned to
Ti 2p _{3/2} (2p _{1/2})	455.04(460.99)	1.10(1.50)	76.98	Ti-C
	458.95 (464.65)	1.47(2.34)	23.02	Ti-O

Figure R2. The EDS mapping of all elements for fig.1d in the manuscript.

Figure R3. The EDS mapping of all elements for samples at Si substrate. Noting that the yellow spectrum is the actual spectrum for Si substrate and the red one is for comparison with the spectrum in figure R4.

Figure R4. The EDS mapping of all elements for samples at Cu matrix substrate. Noting that the Pt element in the spectrum is that sputtering to the sample for better conductivity.

Comments on the XPS results and their analysis:

The XPS analysis in the paper is inadequate and inaccurate, the vast majority of the cited references do not even include XPS data (Refs. 12, 15, 17, 19 and 66). The authors cite reference 67 which is the only reliable reference however they are not following the same calibration (using the fermi level instead of the so-called adventitious carbon and not following the same fitting models with no justification to that. To reliably compare the surface oxidation of Ti and Al between the different samples, a careful analysis must be performed for the XPS data. I would advise the authors to follow the same peak fitting and calibration of Ref. 67 to obtain accurate and reliable data.

Firstly, we apologize for the incorrect citation of references in the manuscript. The accurate citations exclude references 12, 15, 17, and 19; instead, they include only references 65-67 as subsequently cited in the manuscript. In line with your suggestions, we conducted XPS testing on the sample with the highest purity as indicated by XRD results, which was synthesized at 900°C, and analyzed it following the methodology outlined in reference 67. It is noteworthy that although reference 67 reports XPS results for Ti_3AlC_2 MAX phase rather than Ti_2AlC MAX phase, we still followed the asymmetric fitting method from it. Since the primary contention with Reviewer 1 concerns the content of TiO_2 , we have focused our analysis exclusively on the Ti-2p XPS results.

The results demonstrate that our Ti 2p spectrum closely resembles that reported in the literature, with similar peak positions. Furthermore, the ratio of Ti-C to Ti-O obtained from the XPS analysis software approximates 3:1, where the Ti-C part is the majority state of Ti. However, as summarized in the table R1 of the response to reviewer 1, different literature sources exhibit certain deviations in XPS results, with even the peak shapes being inconsistent. This underscores the inherent challenges in analyzing XPS results, and we trust you will appreciate this aspect.

FigureR2. Deconvolution of Ti 2p XPS spectra for 1D- Ti_2AlC samples at reaction temperature of 900 °C.

Table R2. The parameters obtained from Ti-2p curve fitting.

Region	BE (eV)	FWHM (eV)	Fraction	Assigned to
Ti 2p_{3/2}(2p_{1/2})	455.04(460.99)	1.10(1.50)	76.98	Ti-C
	458.95 (464.65)	1.47(2.34)	23.02	Ti-O

References

1. Wilhelmsson, O., Palmquist, J. P., Lewin, E., Emmerlich, J., Eklund, P., Persson, P. Å., ... & Jansson, U. (2006). Deposition and characterization of ternary thin films within the Ti–Al–C system by DC magnetron sputtering. *Journal of crystal growth*, 291(1), 290-300.
2. Mahmoudi, Z., Tabaian, S. H., Rezaie, H. R., Mahboubi, F., & Ghazali, M. J. (2020). Synthesis of Ti₂AlC & Ti₃AlC₂ MAX phases by Arc-PVD using Ti–Al target in C₂H₂/Ar gas mixture and subsequent annealing. *Ceramics International*, 46(4), 4968-4975.
3. Myhra, S., Crossley, J. A. A., & Barsoum, M. W. (2001). Crystal-chemistry of the Ti₃AlC₂ and Ti₄AlN₃ layered carbide/nitride phases—characterization by XPS. *Journal of Physics and Chemistry of Solids*, 62(4), 811-817.
4. Barsoum, M. W., Crossley, A., & Myhra, S. (2002). Crystal-chemistry from XPS analysis of carbide-derived M_{n+1}AX_n (n= 1) nano-laminate compounds. *Journal of Physics and Chemistry of Solids*, 63(11), 2063-2068.
5. Magnuson, M., Wilhelmsson, O., Palmquist, J. P., Jansson, U., Mattesini, M., Li, S., ... & Eriksson, O. (2006). Electronic structure and chemical bonding in Ti₂AlC investigated by soft x-ray emission spectroscopy. *Physical Review B*, 74(19), 195108.
6. Zhang, Z., Lim, S. H., Lai, D. M. Y., Tan, S. Y., Koh, X. Q., Chai, J., ... & Pan, J. S. (2017). Probing the oxidation behavior of Ti₂AlC MAX phase powders between 200 and 1000° C. *Journal of the European Ceramic Society*, 37(1), 43-51.
7. Näslund, L. Å., Persson, P. O., & Rosen, J. (2020). X-ray Photoelectron Spectroscopy of Ti₃AlC₂, Ti₃C₂T_z, and TiC Provides Evidence for the Electrostatic Interaction between Laminated Layers in MAX-Phase Materials. *The Journal of Physical Chemistry C*, 124(50), 27732-27742.
8. Gutzmann, H., Gärtner, F., Höche, D., Blawert, C., & Klassen, T. (2013). Cold spraying of Ti₂AlC MAX-phase coatings. *Journal of thermal spray technology*, 22, 406-412.
9. Chastain, J., & King Jr, R. C. (1992). Handbook of X-ray photoelectron spectroscopy. Perkin-Elmer Corporation, 40, 221.

REVIEWER COMMENTS

Reviewer #1 (Remarks to the Author):

Paper can be published now.

Reviewer #3 (Remarks to the Author):

The authors have improved the fitting significantly. However, I question the utilization of the XPS data to draw the conclusions stated by the authors.

1. In page 5 paragraph 10, the authors indicate that the XPS results validate the purity of 1D-Ti₂AlC with the reaction temperature 900 °C, it is difficult to differentiate between the BE peaks of Ti-C and Ti-C in MAX phase, might be done if reference samples were measured using the same instrument same calibration ect, the same goes for Ti-Al. I recommend removing this claim and rely on XRD since also it gives a macroscopic quantification and covers a large sample size.
 2. In the XPS analysis section page 25 paragraph 40, the the authors indicate that Al exhibits sensitivity to oxygen at elevated temperature at elevated temperatures is that compared to RT samples if so a comparison with a sample at RT is required. Since generally MAX phases at RT exhibit Al-O layer of several nano meters thick.
 3. Looking at Figure S20, the fitting presented in the Al 2p region seems inaccurate the two peaks assigned to Ti-Al species appears to not show a 2:1 areal ratio. I recommend that the authors check all the doublet peaks and make sure they follow the physics laws. Also in the fitting of C 1s, the BE of the C-Ti species is clearly not in the same position as the original data, I recommend the removal of the fitting and simply indicate which peak belongs to which species. Furthermore, the peak assignment of the Ti-O and Al-O does not follow any of the cited papers, ref 65 indicates that both have the same BE, ref 66 does not show the O 1s region, and ref 67 show TiO₂ at 530.5 eV vs. the authors' assignment at 529.9 and Al₂O₃ at 532.8 vs. the authors' assignment at 530.9 eV, also here I recommend the removal of the data on O 1s region since it is inaccurate and does not serve much the paper.
- I recommend the acceptance of the manuscript after the implementation of the points above.

RESPONSE TO REVIEWERS' COMMENTS

Reviewer #3 (Remarks to the Author):

The authors have improved the fitting significantly. However, I question the utilization of the XPS data to draw the conclusions stated by the authors.

1. In page 5 paragraph 10, the authors indicate that the XPS results validate the purity of 1D-Ti₂AlC with the reaction temperature 900 °C, it is difficult to differentiate between the BE peaks of Ti-C and Ti-C in MAX phase, might be done if reference samples were measured using the same instrument same calibration ect, the same goes for Ti-Al. I recommend removing this claim and rely on XRD since also it gives a macroscopic quantification and covers a large sample size.

Thank you for your valuable suggestions. We have revised the manuscript accordingly, replacing the previous statement with that the surface atomic analysis was illustrated in the corresponding figures and tables by XPS results. The discussion of the purity of 1D-Ti₂AlC was confirmed through refined XRD analysis in the second paragraph on page 3 of the revised manuscript.

2. In the XPS analysis section page 25 paragraph 40, the the authors indicate that Al exhibits sensitivity to oxygen at elevated temperature at elevated temperatures is that compared to RT samples if so a comparison with a sample at RT is required. Since generally MAX phases at RT exhibit Al-O layer of several nano meters thick.

We regret any confusion caused by the previous statement regarding the increased affinity of Al for oxygen at elevated temperatures. Our intention was to convey that during the synthesis of 1D-Ti₂AlC, especially under high-temperature conditions and in cases of incomplete reactions, there is an increased propensity for aluminum oxide formation (Fig. S21 and S26). As you correctly noted, an Al-O layer also forms in MAX phases at room temperature (Ref. 65). We have revised the statement to more precisely reflect this understanding.

References

65. Zhang, Z., Lim, S. H., Lai, D. M. Y., Tan, S. Y., Koh, X. Q., Chai, J., ... & Pan, J. S. (2017). Probing the oxidation behavior of Ti₂AlC MAX phase powders between 200 and 1000° C. *Journal of the European Ceramic Society*, 37(1), 43-51.

3. Looking at Figure S20, the fitting presented in the Al 2p region seems inaccurate the two peaks assigned to Ti-Al species appears to not show a 2:1 areal ratio. I recommend that the authors check all the doublet peaks and make sure they follow the physics laws. Also in the fitting of C 1s, the BE of the C-Ti species is clearly not in the same position as the original data, I recommend the removal of the fitting and simply indicate which peak belongs to which species. Furthermore, the peak assignment of the Ti-O and Al-O does not follow any of the cited papers, ref 65 indicates that both have the same BE, ref 66 does not show the O 1s region, and ref 67 show TiO₂ at 530.5 eV vs. the authors' assignment at 529.9 and Al₂O₃ at 532.8 vs. the authors' assignment at 530.9 eV, also here I recommend the removal of the data on O 1s region since it is inaccurate and does not serve much the paper.

I recommend the acceptance of the manuscript after the implementation of the points above.

We apologize for the incorrect fitting of the XPS results, and we have checked the XPS results with new fitting curves for Al-XPS. For the C 1s XPS results, we removed the fitting curves, and pointed the corresponding species. Lastly, we have removed the data of the O 1s region as your request.

FigureR1. Fitting curves of Al 2p XPS spectra for 1D-Ti₂AlC samples at reaction temperature of 900 °C.

FigureR2. C 1s XPS spectra for 1D-Ti₂AlC samples at reaction temperature of 900 °C.